# Brain-to-gut trafficking of alpha-synuclein by CD11c+ cells in a mouse model of Parkinson's disease

Rhonda L. McFleder [1], Anastasiia Makhotkina[1], Janos Groh [2], Ursula Keber [3], Fabian Imdahl[4], Josefina Peña Mosca[5], Alina Peteranderl[1], Jingjing Wu[1], Sawako Tabuchi[1], Jan Hoffmann[1], Ann-Kathrin Karl[1], Axel Pagenstecher[3], Jörg Vogel [4], Andreas Beilhack [5], James B. Koprich[6,7], Jonathan M. Brotchie [6,7], Antoine-Emmanuel Saliba [4,8], Jens Volkmann[1] & Chi Wang Ip [1]✉

Inflammation in the brain and gut is a critical component of several neurological diseases, such as Parkinson's disease (PD). One trigger of the immune system in PD is aggregation of the pre-synaptic protein, α-synuclein (αSyn). Understanding the mechanism of propagation of αSyn aggregates is essential to developing disease-modifying therapeutics. Using a brain-first mouse model of PD, we demonstrate αSyn trafficking from the brain to the ileum of male mice. Immunohistochemistry revealed that the ileal αSyn aggregations are contained within CD11c+ cells. Using single-cell RNA sequencing, we demonstrate that ileal CD11c+ cells are microglia-like and the same subtype of cells is activated in the brain and ileum of PD mice. Moreover, by utilizing mice expressing the photo-convertible protein, Dendra2, we show that CD11c+ cells traffic from the brain to the ileum. Together these data provide a mechanism of αSyn trafficking between the brain and gut.

Parkinson's disease (PD)[1] and multiple sclerosis (MS)[2] are among the many neurological diseases that exhibit neuroinflammation, which contributes to disease development and progression. Interestingly, inflammation is not confined to the central nervous system (CNS), the gut also exhibits signs of inflammation in these neurological disorders[1-3]. This gut inflammation is associated with microbiotia changes and gastrointestinal (GI) complications[4], which in the case of PD, can occur prior to the development of motor dysfunction[5]. As gut inflammation may precede neuroinflammation, speculations have arisen that inflammation in the gut causes the neuroinflammation[1,4]. Indeed, in MS, gut-derived T cells were found to infiltrate the brain,

suggesting a direct role for these gut-derived immune cells in neuroinflammation[2]. Although activation of the immune system in the gut is clear, the trigger for this inflammation remains uncertain. Identifying the origin and trigger of gut inflammation in neurological diseases is critical to understanding these multi-system interactions and thus for the development of disease-modifying therapies targeting neuroinflammation.

While αSyn is physiologically needed for normal immune function[6], aggregated αSyn is known to trigger an immune response in PD[1,7]. Interestingly, αSyn aggregates are not only found in the brain of PD patients but also in the gut, where accumulations can be identified

[1]Department of Neurology, University Hospital of Würzburg, Würzburg, Germany. [2]Institute of Neuronal Cell Biology, Technical University Munich, Munich, Germany. [3]Department of Neuropathology, Philipps University of Marburg, Marburg, Germany. [4]Helmholtz Institute for RNA-based Infection Research (HIRI), Helmholtz-Center for Infection Research (HZI), Würzburg, Germany. [5]Department of Internal Medicine II, Center for Experimental Molecular Medicine (ZEMM), Würzburg University Hospital, Würzburg, Germany. [6]Atuka Inc., Toronto, ON, Canada. [7]Krembil Research Institute, Toronto Western Hospital, University Health Network, Toronto, ON, Canada. [8]Faculty of Medicine, Institute of Molecular Infection Biology (IMIB), University of Würzburg, Josef-Schneider-Str. 2, 97080 Würzburg, Germany. ✉e-mail: ip_c@ukw.de

even before the onset of motor symptoms[5]. Although neuron to neuron propagation of αSyn has been proposed as the likely method of αSyn transport throughout the body, it does not seem to be true for every patient. In fact, a large majority of patients do not fit this classical route, suggesting that αSyn and thereby PD may propagate via additional mechanisms[5]. Both monomeric and aggregated forms of αSyn exhibit chemoattractant properties, and have been shown to recruit both human and mouse monocyte-derived myeloid cells and microglia cells[8,9]. Recruited monocyte-derived myeloid cells, such as macrophages and dendritic cells, can engulf αSyn and consequently activate the adaptive immune cells responsible for the αSyn-specific inflammation that occurs in PD[10,11].

As phagocytic cells are activated by αSyn aggregates and capable of traveling throughout the body[9], we hypothesized that perhaps these cells could facilitate αSyn trafficking. To test this hypothesis, we leveraged a mouse model of PD in which an adeno-associated virus (AAV) encoding the human A53T-mutated form of αSyn (hαSyn) is injected directly into the substantia nigra (SN) of mice[12]. Because the disease is initiated in the brain, this model represents a brain-first model that develops both neuroinflammation and neurodegeneration over the course of 10 weeks[11]. In contrast to other brain-first models, this model also develops αSyn aggregations in the brain making it an ideal model for our study. Here we demonstrate that despite confinement of the virus to the brain, hαSyn could be detected in the ileum of PD mice. This ileal-localized αSyn formed insoluble aggregates, similar to those found in the brain, and resulted in GI dysfunction. Immunohistochemistry and flow cytometry demonstrated that ileal-αSyn was sequestered within CD11c[+] cells. Single-cell RNA sequencing (scRNA-seq) revealed that the brain and ileum share a distinctive subpopulation of CD11c[+] microglia-like cells that become activated in PD animals, in addition to a unique subset of migrating macrophages. By combining optic fiber implantation and mice expressing the photoconvertible protein, Dendra2, we could definitively demonstrate that CD11c[+] cells exit the brain and travel to the ileum. These findings suggest a unique communication between the brain and the ileum, mediated by CD11c[+] cells. The identification of the CD11c[+] cells involved in brain–gut communication could provide a useful target for halting αSyn propagation and PD progression.

## Results

### Over-expression of pathological αSyn in the SN leads to αSyn accumulation in the ileum and gastrointestinal dysfunction

To test if αSyn can propagate from the SN to the intestines, an AAV encoding the hαSyn or an empty vector control (EV) was injected unilaterally into the SN of WT mice. This brain-first PD model preferentially induces localized expression of hαSyn in neurons[13] (Fig. 1a), which is associated with dopaminergic neurodegeneration 5 weeks following the injection (Fig. 1a–c). At the same time point, αSyn[+] cells could also be detected in the lamina propria of the distal ileum (Fig. 1d, e). Although most of the ileal αSyn was endogenous, the hαSyn was also present in the ileum and could be distinguished by the presence of a hemagglutinin (HA) tag present at the C-terminus (Fig. 1a, d, f and Supplementary Fig. 1a). Of note, αSyn in the lamina propria was not specific for the viral vector model of PD but could also be shown in the A30P/A53T transgenic model of PD (Supplementary Fig. 1b). In contrast, αSyn knockout mice (SNCAKO) exhibited no αSyn staining (Supplementary Fig. 1c). Interestingly, αSyn could not be detected in other organs such as spleen or colon (Fig. 1g, h), indicating a deliberate trafficking of the protein between the brain and ileum. Brain-gut trafficking was preferential to αSyn, as expression of green fluorescent protein (GFP) in the SN lead to only minimal amounts of GFP in the ileum (Fig. 1a, d, i). Similar to other brain-first PD mouse models[14], the hαSyn PD mice also exhibited increased fecal pellet output with no change in stool water content (Supplementary Fig. 1d, e). Whole gut transit was also measured and found to be decreased in comparison to

controls (Fig. 1j). This perturbation in gastrointestinal function, demonstrates the pathophysiological impact of αSyn protein accumulations.

In PD, it is not the normal αSyn but rather its pathologic forms that are thought to be detrimental to dopaminergic neurons[15]. To determine if ileal αSyn was pathologically altered, we measured two pathologic forms: aggregated and phosphorylated αSyn (pαSyn). Aggregated αSyn was defined as αSyn molecules that were resistant to proteinase K (PK) treatment. In the SN of hαSyn PD mice, aggregated αSyn slowly accumulated over the course of 10 weeks (Fig. 2a, b and Supplementary Fig. 1g). This was in contrast to tyrosine hydroxylase (TH), a marker of dopaminergic neurons, which did not aggregate and was completely digested by PK-treatment (Supplementary Fig. 1g). Both endogenous αSyn and virally expressed hαSyn aggregated in the ileum of hαSyn mice, where it was surprisingly already accumulating one week post-injection (Fig. 2a, c). pαSyn was also significantly elevated early on in the ileum of hαSyn animals (Fig. 2d, e), whereas in the SN it could only be detected at 10 weeks after injection (Supplementary Fig. 1f). These data demonstrate that in our brain-first model of PD, pathological forms of αSyn are apparent at early timepoints in the intestines. Interestingly, in both the SN and the ileum, most pathologic αSyn molecules were HA negative. This suggests that protein aggregations in the brains of hαSyn animals consist mainly of endogenous αSyn, likely explaining why mostly endogenous αSyn propagates to the ileum in this model.

### CD11c[+] cells engulf αSyn in the brain and the ileum of PD mice

The data above (Figs. 1 and 2) demonstrate propagation of αSyn from the brain to the ileum, however, the mechanism underlying this transport remained unclear. As αSyn has been shown to propagate via the vagus nerve in other models of PD[16], we next investigated the vagus nerve of hαSyn mice. Staining of the dorsal motor nuclei of the vagus nerve for αSyn and HA in hαSyn mice 1 and 5 weeks post-injection did not demonstrate accumulation of either of these proteins at the two different time points. The vagus nerve was also stained at the 5 week time point and likewise lacked αSyn and HA signal (Supplementary Fig. 2a, b). These data indicate that αSyn is not propagating via the vagus nerve route from brain to gut in this mouse model.

The ileal αSyn accumulations were surprisingly localized within the *lamina propria*, an area enriched with various immune cells[17] (Fig. 1d and Supplementary Fig. 2c). Indeed, pαSyn but not GFP protein could be detected in the circulating white blood cells (WBC) from hαSyn or GFP-injected animals (Supplementary Fig. 2d–f). We next tested if WBCs could be carrying αSyn by performing co-staining for αSyn in addition to immune cell markers at the 5 week time point. Intriguingly, αSyn tended to colocalize with CD11c[+] cells in the ileum (Fig. 3a, top panel). These CD11c[+] αSyn[+] cells could be detected by both immunofluorescence and flow cytometry analysis of the ilea of hαSyn animals (Fig. 3a–c and Supplementary Fig. 2g). CD11c[+] cells were also present in the SN of hαSyn mice, where they were similarly positive for αSyn (Fig. 3a). Likewise, every SN analyzed from PD patient brain autopsies demonstrated CD11c[+] αSyn[+] cells. This was in stark contrast to non-PD patients, where no CD11c[+] αSyn[+] cells were detected in the SN (Fig. 3d, e).

Brain and ileal-localized CD11c[+] cells were not just αSyn positive, but they also showed increased activation in the hαSyn animals. Using flow cytometry, significant increases in CD86 expression could be measured in the CD11c[+] cells present in both the brain and the ileum of hαSyn animals 10 weeks post-injection (Fig. 3f, g). This was in stark contrast to the spleen which did not show a difference at any of the time points measured. Interestingly, the activated cells were negative for CD103, a marker for dendritic cells, indicating that they may represent CD11c[+] macrophages (Supplementary Fig. 2h). CD11c[+]αSyn[+] cells could also be identified in the terminal ileum of a PD patient (Fig. 3h), indicating the translational significance of these results.

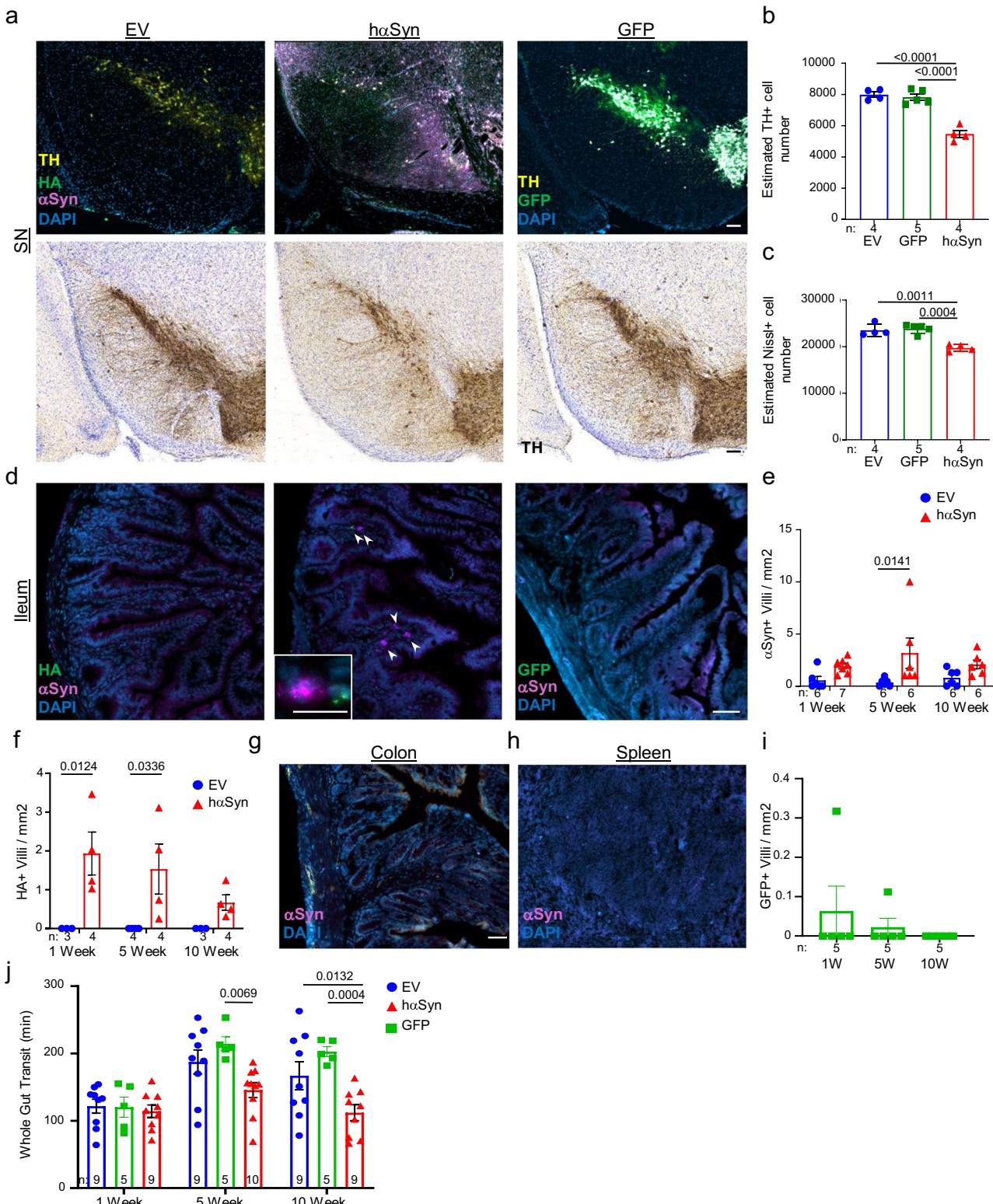

To test if CD11c[+] cells mediate the propagation of αSyn in the hαSyn mouse model, we utilized the transgenic B6a.CD11c.DOG mouse strain. These mice express the diphtheria toxin receptor under the CD11c promoter, allowing for prolonged depletion of CD11c[+] cells with multiple diphtheria toxin injections[18]. One week after stereotactic injection with an AAV expressing hαSyn, WT and transgenic B6a.CD11c.DOG littermates received IP injections of diphtheria toxin every other day for two weeks. At week three after injection, the intestines of these mice were analyzed for αSyn. The 3-week timepoint was used, as previous studies have demonstrated that CD11c can only be depleted for two weeks with this mouse strain. Transgenic mice with depleted CD11c demonstrated a notable yet insignificant (p value 0.087) reduction in the number of αSyn[+] villi compared to their WT controls (Fig. 3i and Supplementary Fig. 2i). Taken together these data demonstrate a unique connection in PD mice between the brain and the ileum, mediated by CD11c[+] macrophages.

**Fig. 1 | hαSyn expression in the brain leads to increased αSyn in the ileum of PD mice.** All images are from mice five weeks following stereotactic injection of the AAV vector. **a** Top panel: Representative immunofluorescence images for Tyrosine Hydroxylase (TH) (yellow), HA (green), αSyn (magenta), and DAPI (blue) in the SN of either EV or hαSyn mice. Far right demonstrates a representative immuno-fluorescence image of GFP (green), TH (yellow), and DAPI (blue) in the SN of mice injected with an AAV-GFP. Bottom Panel: Representative immunohistochemistry images of TH. **b** Quantification of TH positive or Nissl-stained neurons (**c**) in the SN. **d** Representative immunofluorescence images of HA (green) and αSyn (magenta) in the ileum of EV, hαSyn, and GFP mice. Arrowheads indicate αSyn positive cells.

**e** Quantification of the number of αSyn positive villi in the ileum of EV or hαSyn mice. **f** Quantification of the number of HA positive villi in the ileum of hαSyn mice. **g** Representative immunofluorescence images of either the colon or the spleen (**h**) of hαSyn mice ($n = 3$). **i** Quantification of GFP positive villi in the ileum of GFP mice. **j** Whole gut transit quantification at 1, 5, and 10 weeks after injection in EV, hαSyn, or GFP mice. Scale bars represent 50 μm in larger images and 10 μm in the inserts. All graphs depict EV (blue), hαSyn (red), and GFP (green). Statistical analysis by one-way ANOVA with Tukey's post hoc (**b**, **c**, **i**) or two-way ANOVA with Bonferroni's post hoc test (**e**, **f**, **j**). Data are presented as mean values +/− SEM. Source data are provided as a Source data file.

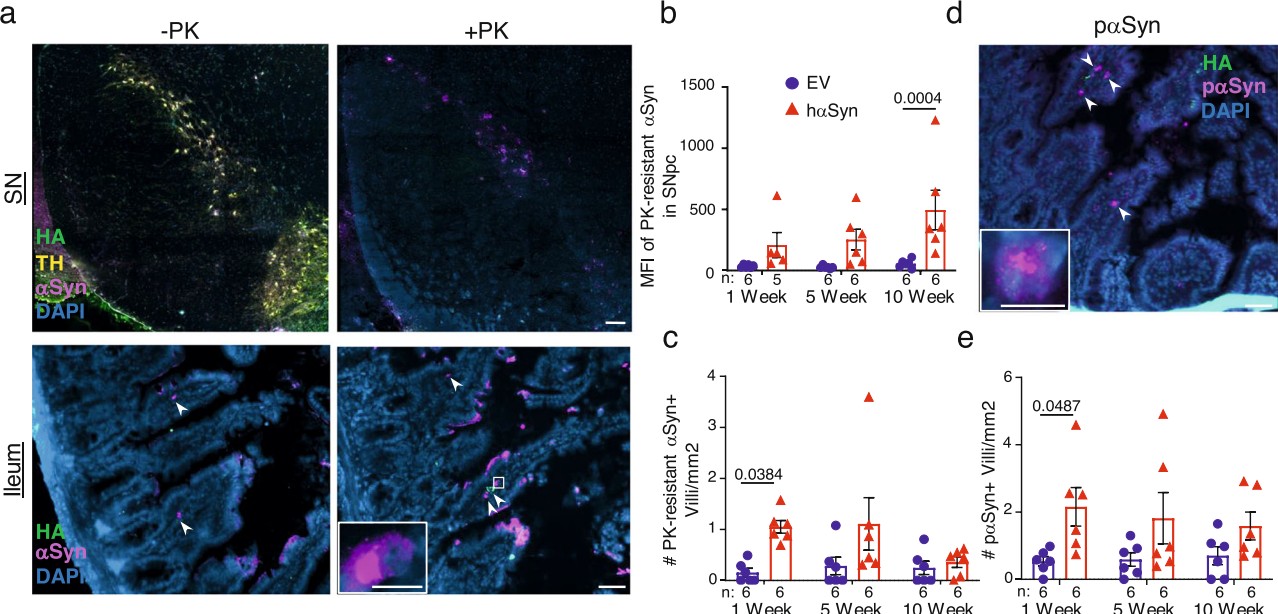

**Fig. 2 | Pathological αSyn accumulates in PD mice ilea. a** Representative immunofluorescence images of TH (yellow), αSyn (magenta), and DAPI (blue) without (−PK) or with (+PK) Proteinase K (PK) treatment in the SN (top) or the ileum (bottom) of hαSyn mice 5 weeks following the injection. **b** Quantification of the MFI of αSyn in the +PK SN of EV and hαSyn mice at 1, 5, and 10 weeks following AAV injection. **c** Quantification of the αSyn⁺ profiles in villi following PK treatment in the ileum of EV or hαSyn animals at 1, 5, and 10 weeks following viral vector injection. **d** Representative image of HA (green), pαSyn (magenta), and DAPI (blue) in the

ileum of an hαSyn animal five weeks following viral vector injection. **e** Quantification of the number of pαSyn⁺ villi in the ileum of the EV and hαSyn animals at 1, 5, and 10 weeks following viral vector injection. *N* numbers are indicated below each graph. All graphs depict EV (blue) and hαSyn (red). Scale bars represent 50 μm in larger images and 10 μm in the insert. Statistical analysis by two-way ANOVA with Bonferroni's post hoc test. Data are presented as mean values +/− SEM. Source data are provided as a Source Data file.

## The ileum contains microglia-like CD11c⁺ cells

To further elucidate the shared role of CD11c⁺ cells in the brain and ileum, droplet-based 10x Genomics scRNA-seq was performed on freshly isolated CD11c⁺ cells from the brain, spleen and ileum from EV and hαSyn mice 5 weeks post-injection (see Methods). The five week time point was chosen due to the large number of αSyn⁺ cells present in the ileum at this stage. A total of 5 mice from each group were pooled and CD45⁺CD11c⁺ cells from each organ were sorted and sub-jected to scRNA-seq (Supplementary Fig. 2j). After filtering out low-quality cells, 5956 cells could be partitioned into 12 distinct CD11c⁺ clusters which were then further annotated using the Mygeneset tool in the Immunological Genome Project[19] (Fig. 4a, b and Supplementary Data 1). Two distinct tissue resident macrophage (TRM) clusters could be identified, TRM 1 and TRM 2. These clusters were enriched with classical microglia markers such as *Cx3cr1* and *P2ry12* and indeed 81.78% of brain CD11c⁺ cells fell into one of these two clusters (Fig. 4c and Supplementary Fig. 3a). Four distinct dendritic cell clusters were also identified and were enriched for MHC2 genes such as *H2-Eb1* and *H2-Ab1* (Fig. 4a, b and Supplementary Fig. 3b). Interestingly the Den-dritic 1 cluster was also enriched for *Lrrk2*, a gene associated with PD[20] (Supplementary Fig. 3c). Four macrophage populations could also be

identified in addition to two monocytic populations (Fig. 4a, b and Supplementary Fig. 3d, e). Importantly, none of these cells expressed significant levels of αSyn mRNA (*Snca*) (Supplementary Fig. 3f). The *hαSyn* gene was also examined, where a few mapped reads were detected. However, these reads did not signify expression of the *hαSyn* in the CD11c⁺ cells analyzed, as they were present in low quality cells and filtered out after the first quality control step of our analysis pipeline (Supplementary Fig. 3g, h). These data provide further evidence that the αSyn in the ileum was not a result of increased ileal αSyn expression but rather transport of protein aggregates from the brain.

Analysis of the distribution of the cells from the different organs revealed that brain and spleen CD11c⁺ cells clustered away from each other and tended to form their own distinct clusters, whereas the ileal cells seemed to be more equally distributed (Fig. 4c). Remarkably, only 0.64% of splenic CD11c⁺ cells clustered into the microglia-like TRM 1 cluster, which was in stark contrast to the 8.33% of ileal CD11c⁺ cells (Fig. 4c). As the cells in the TRM 1 cluster express classical microglia-specific markers such as *Sall1* and *P2ry12* (Supplementary Fig. 3a, b), ileal CD11c⁺ cells present in this cluster represent a microglia-like TRM population. This surprising finding was further supported by immunofluorescence and western blot staining against the microglia

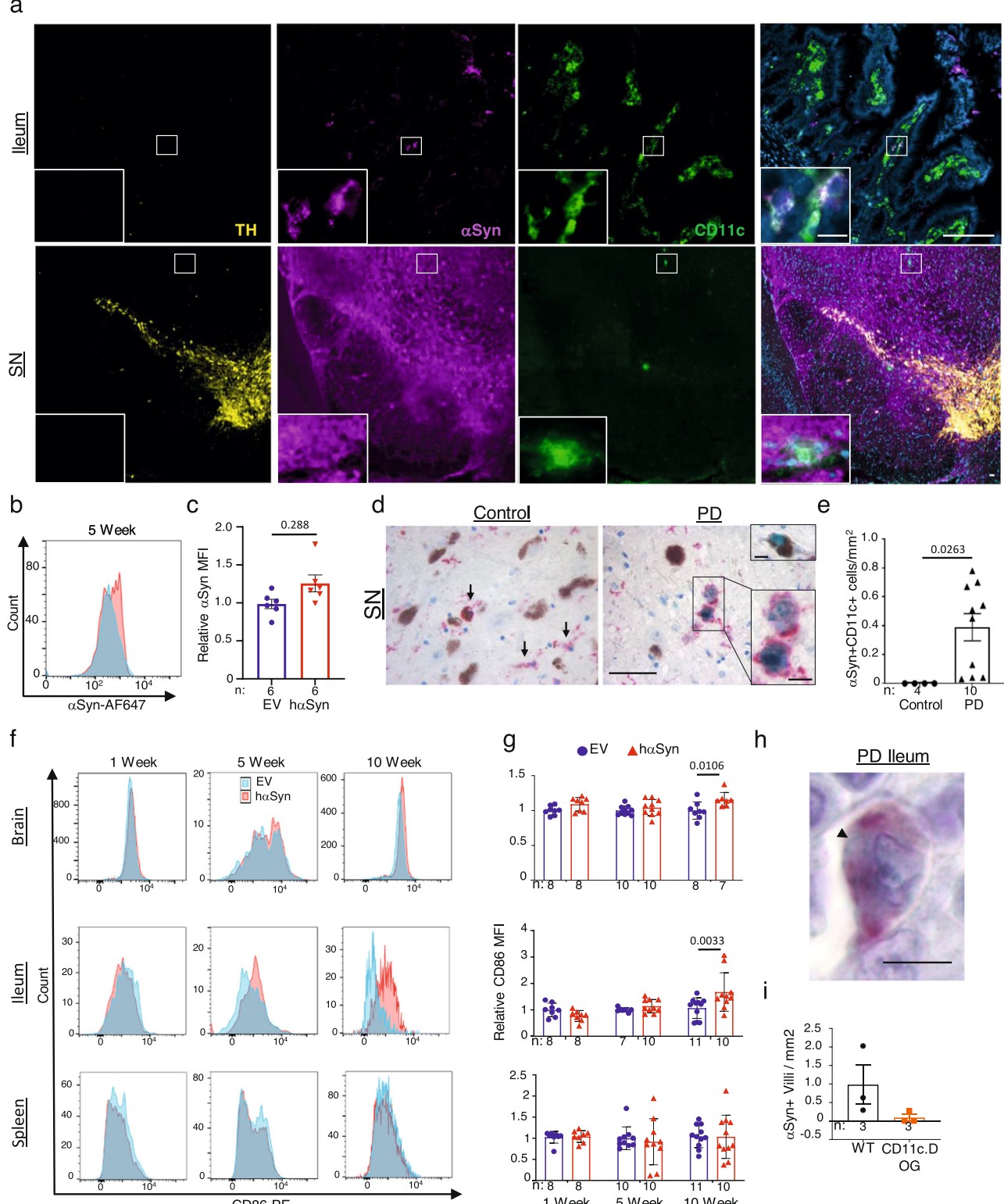

**Fig. 3 | CD11c⁺ aSyn⁺ cells are present in the brain and ileum of PD mice and patients. a** Representative immunofluorescence images of TH (yellow), αSyn (magenta), CD11c (green), and DAPI (blue) from the ileum (top) and the SN (bottom) of hαSyn mice 5 weeks following viral vector injection (representative of 5 mice). Representative histogram plot (**b**) and quantification (**c**) of αSyn MFI in the ileal CD11c⁺ cells of hαSyn (red) or EV (blue) mice 5 weeks following viral vector injection. **d** Representative images and quantification (**e**) of CD11c (red) and αSyn (green) in the SN of control or PD patients. Arrows represent CD11c⁺ cells, zoomed in image of αSyn⁺CD11c⁺ cells, insert depicts a neuron containing an αSyn⁺ Lewy body. **f** Representative histogram plots and quantification (**g**) of CD86 expression in CD11c⁺ cells. **h** Representative image of CD11c (red) and αSyn (green) from the terminal ileum of one PD patient. Arrowhead indicates αSyn⁺ cell. **i** Quantification of aSyn⁺ Villi in hαSyn WT (black) and CD11c.DOG (orange) mice treated with diphtheria toxin. Statistical analysis by unpaired one-tailed Student's *t* test (**c**, **e**, **i**) and two-way ANOVA with Bonferroni's post hoc test (**g**). Scale bars represent 50 μm in larger images and 10 μm in the inserts for (**a**, **d**), and 20 μm for **h**. Data are presented as mean values +/− SEM. Source data are provided as a Source data file.

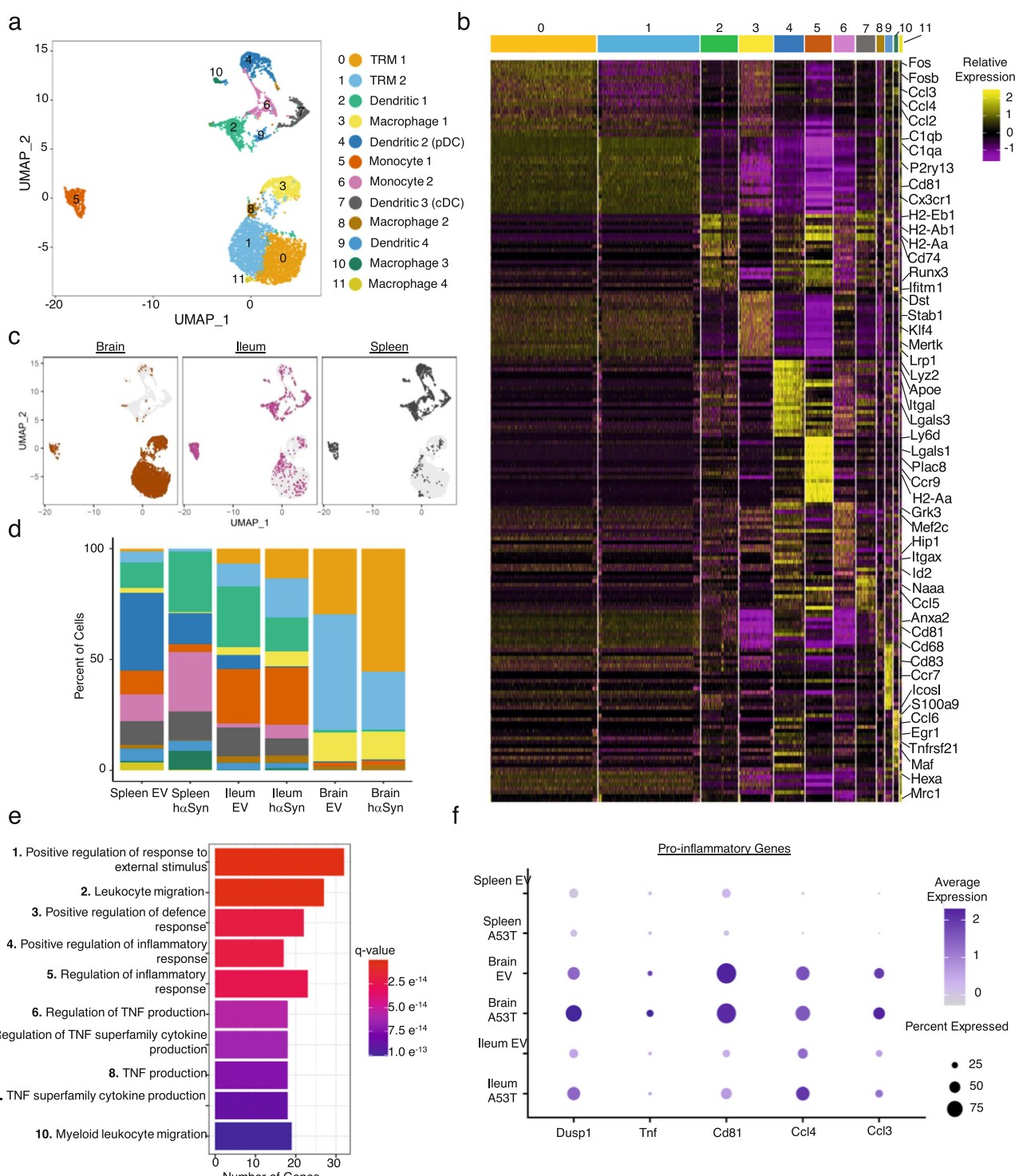

**Fig. 4 | The ileum and brain share a unique microglia-like CD11c+ population.** **a** UMAP of the different CD11c+ single-cell transcriptome subpopulations identified in the spleen, brain, and ileum from EV and hαSyn animals. **b** Heatmap demonstrating the top 20 markers identified in each CD11c+ cluster. Clusters are color-coded along the top of the graph using the same colors in (**a**). **c** UMAP as in (**a**) demonstrating the cells partitioned by cellular organ of origin: distinct clustering of the CD11c+ cells from brain (brown), ileum (magenta), or spleen (gray) of EV and

hαSyn animals. **d** Bar graph displaying the relative percent of each CD11c+ cluster in the spleen, ileum, and brain of EV or hαSyn animals. **e** Bar graph of the top ten upregulated GO terms enriched in the TRM 1 cluster. **f** Dotplot demonstrating the average expression of select pro-inflammatory genes present in the top 5 GO Terms described in (**e**). Sequencing data is from the pooled organs of 5 EV and 5 hαSyn animals harvested 5 weeks following viral vector injection.

marker, P2RY12, in addition to CD11c in the SN and the ileum of hαSyn mice (Supplementary Fig. 4a, b). In the SN, P2RY12 colocalizes with CD11c+ microglia cells, 5 weeks following injection (Supplementary Fig. 4a). This is also true in the ileum, where most of the P2RY12

colocalizes with CD11b, however some CD11c, CD11b, P2RY12 triple positive cells could also be identified (Supplementary Fig. 4a). These data indicate that the brain and the ileum share a unique TRM population that is transcriptionally similar to microglia.

## Pro-inflammatory microglia-like TRM cells increase in both the brain and ileum of PD mice

To visualize the effect of αSyn accumulation on these CD11c$^+$ populations, the relative proportion of the different clusters was analyzed in the three organs of either EV or hαSyn animals. The proportion of TRM 1 cells increased in both the ileum and the brain of hαSyn animals compared to EV (Fig. 4d). The TRM 2 population frequency was also increased in the ileum of hαSyn mice, however this increase was not apparent in the brains. Consistent with our previous data of systemic immune dysregulation in PD[11], splenic CD11c$^+$ cells also underwent shifts in their cluster distributions. Expansion of TRM 1 cells, however, was specific to the brain and the ileum. By performing Gene Ontology (GO) analysis on the genes enriched in the TRM 1 cluster, it became apparent that this cluster represents a pro-inflammatory microglia-like population (Fig. 4e and Supplementary Data 1). TRM 1 was enriched for terms such as: Positive regulation of inflammatory response, TNF production, and Myeloid leukocyte migration. Closer inspection of the genes present in these GO terms, revealed enrichment of genes previously linked to PD such as: Lag3[21], Hspb1[22], and Cebpa[23] (Fig. 4e and Supplementary Figs. 3c and 4c). A dotplot of selected pro-inflammatory genes in the top five GO terms enriched in the TRM 1 cluster demonstrates a noticeable increase in the expression of these genes in the hαSyn ileum and brains compared to EV. Consistent with our previous data, pro-inflammatory transcriptional changes within CD11c$^+$ cells appeared to be specific to the brain and ileum since the spleens from the same animals showed little to no change (Fig. 4f). Cytokine analysis of bulk protein from the ileum likewise revealed an increased in CCL3 protein levels (Supplementary Fig. 4d). Although the CCL3 in the SN was below the detection limits, CXCL10 was significantly increased (Supplementary Fig. 4e). CXCL10 is classically associated with pro-inflammatory microglia and macrophages[24]. Together these data suggest that hαSyn expression in the brain activates pro-inflammatory microglia-like populations in the brain and the ileum.

## CD11c$^+$ cells traffic from the brain to the ileum

The TRM 1 cluster was not the only population shared between the brain and the gut, the Macrophage 1 population was also enriched in these two organs relative to spleen (Fig. 4a, c, d). This Macrophage 1 population also expressed typical microglia markers such as Sall1, P2ry12, and Tmem119, albeit at a lower level than the microglia in the TRM 1 and 2 clusters (Fig. 5a). In contrast to the microglia-like populations, these cells were CD45-high (Fig. 5b), a marker of infiltrating macrophages in the brain[25]. GO analysis of the genes enriched in this cluster revealed categories related to immune cell migration and tissue migration, suggesting that this brain-gut specific population represents a subset of migrating cells (Fig. 5c, d and Supplementary Data 1). Indeed, six out of the top ten GO terms in the Macrophage 1 cluster were migration-related, which was in sharp contrast to the other CD11c$^+$ clusters which contained a maximum of two migration-related terms (Supplementary Data 1).

In addition to migration-related genes (Fig. 5d), Macrophage 1 was also enriched with the PD-related gene Lrp1 (Supplementary Fig. 4f)[26]. The low-density lipoprotein receptor (LDLR)-related protein 1 (LRP1) was recently demonstrated to be a receptor for αSyn on neurons, mediating its uptake and trafficking[27]. LRP1 is not only expressed on neurons however, it is also expressed on macrophages and microglia and serves as an important regulator of their activation[28]. By performing IHC for LRP1 in addition to HA, we were able to show that around 40% of ileal cells containing HA were also enriched for the αSyn-receptor, LRP1 (Fig. 5e and Supplementary Fig. 4g). Interestingly this was not true at the 10-week timepoint, likely reflecting a spread of the hαSyn to other cell types at later timepoints. Around 80% of the ileal hαSyn-containing cells were also positive for the macrophage-specific marker, F4/80 (Supplementary Fig. 4h). This data further

supports the scRNA-seq data and indicates that the Lrp1-enriched Macrophage 1 population might transport αSyn from the brain to the distal ileum in PD mice.

Our previous results suggest that a CD11c$^+$ macrophage population migrates between the brain and the ileum and thereby mediates αSyn transport. To definitively test if CD11c$^+$ cells can exit the brain and enter the ileum, mice expressing the photoconvertible protein, Dendra2, were injected with the AAV encoding either hαSyn or EV control. Immediately after the injection these mice were implanted with optic fibers directly above the SN pars compacta. The blood brain barrier was allowed to recover for one week before blue light at a wavelength of 473 nm was directed to the SN via the optic fibers. Upon blue light illumination Dendra2 undergoes an irreversible red-shift of both excitation and emission spectra. This localized blue light exposure was repeated once per week for 4 weeks, in order to photoconvert Dendra2 in all cells in the SN from green to red fluorescent forms (Supplementary Fig. 5a). At week five following injection, the mice were sacrificed and the brain, ileum, spleen, cervical lymph node, and blood were examined for the presence of photoconverted red fluorescent cells. Following this protocol an average of 0.2% and 0.1% of the total immune cells in the EV and hαSyn Dendra2 mice brains, respectively, were photoconverted (Fig. 6a, b and Supplementary Fig. 5b). Interestingly, both groups also demonstrated photoconverted cells in the ileum. These data suggest that even in non-disease states, cells traffic out of the brain to the ileum (Fig. 6a, b). The percent of photoconverted cells in the ileum significantly correlated with the photoconverted cells in the brain with an $r^2$ value of 0.8186 (Fig. 6c). Translocation of brain cells was specific to the ileum as neither the spleen nor the cervical lymph nodes contained photoconverted cells (Fig. 6d and Supplementary Fig. 5c). By staining for CD11c, we could further show that the majority of photoconverted immune cells in the SN were CD11c$^+$ (Fig. 6e, f). The same was also true in the ileum, albeit not as dramatic (Fig. 6f). Together these data provide direct evidence of trafficking of CD11c$^+$ cells from the brain to the ileum, providing a likely mechanism of propagation of αSyn between the two organs.

## Discussion

In this study, we demonstrate trafficking of αSyn from the brain to the gut in PD mice via CD11c$^+$ macrophages. While several studies have demonstrated bidirectional propagation of αSyn pathology[29–31], it was not clear if this represented propagation of αSyn molecules or a propagation of disease where the pathologic αSyn in one cell leads to aggregation of endogenous αSyn in the next cell. Through the use of an HA-tag, our study definitively shows that αSyn molecules traffic out of the brain and to the gut. In most of the aforementioned studies, propagation of disease occurred along the vagus nerve, whereas in our model this transit appears vagus-independent. The observed speed of pathologic αSyn translocation from the brain to the gut, supports this assumption since the vagus-dependent transit described in other studies is a slow process taking several months to a year[16,29–31]. It is however possible that at later timepoints αSyn can also undergo neuron to neuron trafficking in this model. In non-human primates[30], propagation of αSyn pathology was also vagus-independent. Although the authors did not identify the cell type responsible for the transport, the αSyn appeared to travel via the blood implicating immune cells as the likely transporters.

When injected into the SN, the AAV viral vectors have been shown to have high specificity for neurons[32]. Thus, the αSyn molecules present in the CD11c$^+$ cells likely represent neuronally expressed αSyn that was released and subsequently engulfed by the CD11c$^+$ macrophages. This mechanism of transfer of αSyn has been described by several publications[33,34]. Although the scRNA-seq data is limited by the lack of a positive cell type expressing hαSyn, the absence of this mRNA in our CD11c$^+$ cell scRNA-seq data supports a transfer of αSyn protein from neurons to CD11c$^+$ cells. Depletion of CD11c$^+$ cells seemed to decrease

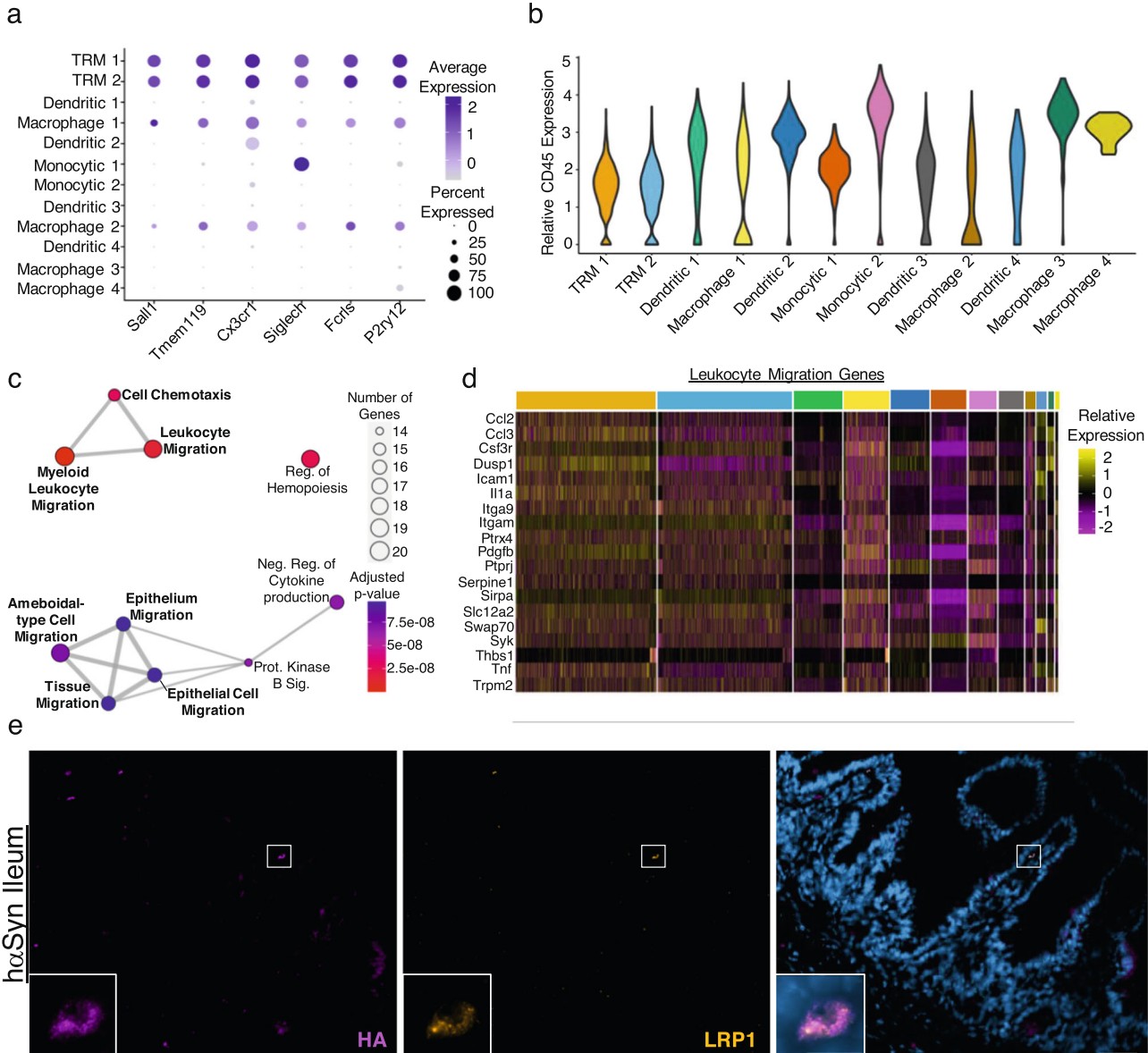

**Fig. 5 | The brain and ileum share a cluster of migrating CD11c⁺ cells. a** Dotplot of selected microglia marker genes. **b** Violin plot of *Ptprc* (CD45) expression in the different CD11c⁺ clusters. **c** Network plot of the top 120 upregulated GO terms in the Macrophage 1 cluster. Migration related terms are bolded. Enrichment was determined using a one-sided hypergeometric test and the Benjamini–Hochberg correction for multiple comparisons. **d** Heatmap of the genes in the Leukocyte Migration GO term. **e** Representative IHC of LRP1 and HA in the distal ileum of hαSyn mice five weeks post-OP (*n* = 3 mice). Scale bars represent 50 μm in the larger images and 10 μm in the inserts.

transport of αSyn to the ileum of hαSyn mice, however this finding is limited by the low n number and transient decrease in CD11c. Future studies which invoke long-term depletion of CD11c⁺ cells through the use of CD11c-cre animals are required to definitively demonstrate that CD11c⁺ cells are required for αSyn transport. αSyn-containing CD11c⁺ cells were also found in PD patients (Fig. 3d, e, h), signifying the clinical relevance of these findings.

The hαSyn mice unexpectedly exhibited increased fecal pellet output and decreased whole gut transit. This is in contrast to other mouse models of PD that involve the enteric neurons and exhibit decreased fecal pellet output[35,36]. This is likely a result of the inflammation occurring in the ileum at this time and could represent an early phase of disease before the onset of constipation resulting from the involvement of the enteric neurons. These results are similar to what was found in the MitoPark mouse model, which also exhibits intestinal inflammation and decreased whole gut transit time[37].

Despite being a brain-first model of PD, the hαSyn animals demonstrated αSyn aggregations in the ileum at early stages of disease. This is in line with patient data describing both gastrointestinal symptoms such as obstipation[38] and αSyn accumulations in the gut prior to the development of motor symptoms[39]. These data demonstrate that the early presence of αSyn aggregation in the gut does not necessarily signify that the disease originated there, as previously thought.

Here we show that the brain and the ileum share a unique subset of microglia-like cells that is not present in the spleen. This is in line with previous microarray data demonstrating that intestinal macrophages express several microglia markers such as: *Cx3cr1*, *Fcrls*, and *P2ry12*[40]. This was in contrast to macrophages from the spleen, lung, peritoneum, and the bone marrow which expressed little to no microglia-specific markers[40]. Tissue resident macrophages may not be the only immune cell type shared between the brain and the gut. A

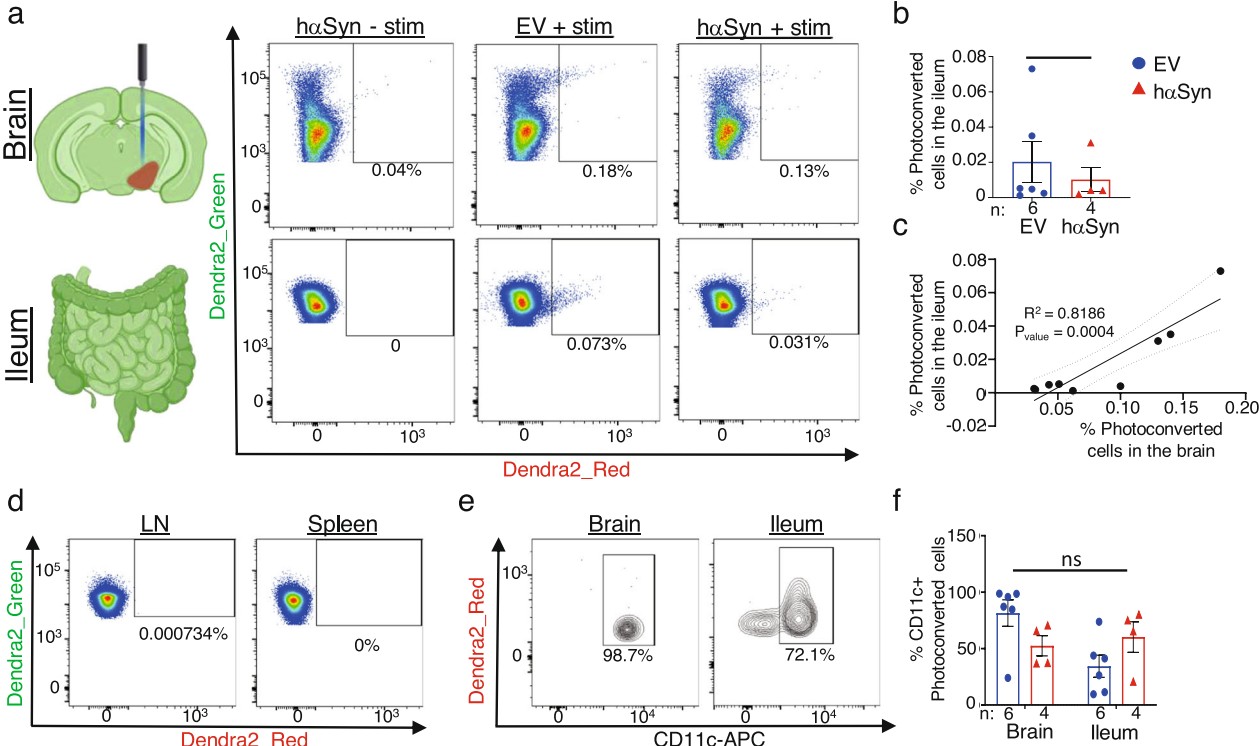

**Fig. 6 | CD11c⁺ cells traffic from the brain to the ileum. a** Diagram (created with BioRender.com) and representative dotplots of photoconverted cells in the brain (top) and the ileum (bottom) from either hαSyn or EV with photoexposure (+stim) or without photoexposure (−stim). **b** Quantification of the % of photoconverted Dendra2 cells in the ileum. **c** Linear regression analysis of the percent of photoconverted cells in the ileum versus the photoconverted cells in the brain ($n = 8$). **d** Representative FACs analysis of Dendra2_Green and Dendra2_Red cells from the cervical lymph node and spleen of photoconverted animals ($n = 8$), images shown are from the EV + stim animal in (**a**). **e** Representative density plots and quantification (**f**) of the percent CD11c⁺ cells in the brain and ileum of photoconverted mice. All graphs depict EV (blue) and hαSyn (red). Statistical analysis by unpaired one-tailed Student's $t$ test (**b**) and two-way ANOVA with Bonferroni's post hoc test (**f**). Data are presented as mean values +/− SEM. Source data are provided as a Source data file.

recent publication, demonstrated the presence of brain-specific regulatory T cells (Tregs) that infiltrate the brain in a stroke mouse model. These brain-specific Tregs were transcriptionally distinct from Tregs isolated from the spleen, pancreas, and adipose tissue, however showed a striking similarity to those isolated from the colon[41]. Why the intestinal tissue resident immune cells are brain-like is unclear, however it is postulated to be due to the neuronal signals from the enteric nervous system[40].

In this PD model, a pro-inflammatory microglia-like TRM population increased in both the brain and the intestines of animals. This pro-inflammatory population was enriched for the gene *Lag3* that is associated with the development of PD[21]. LAG3 was shown to preferentially bind aggregated αSyn in vitro and mediate endocytosis and cytotoxicity in cortical neuronal cultures[42]. It could therefore be the molecule mediating αSyn-induced activation of the microglia-like TRM cells. Indeed deletion of *Lag3* in a transgenic mouse model of PD resulted in decreases in both αSyn aggregates and microglia activation[43]. While LAG3 is normally thought of as an immune checkpoint molecule[44], suppressing immune activation, it will be important to study if perhaps it could have other pro-inflammatory effects in the presence of αSyn aggregates.

By utilizing a Dendra2 transgenic mouse, we were able to track cells trafficking out of the brain and to the ileum of mice. A similar method was used in a mouse model of MS, where CD4⁺ cells were found to travel from either the inguinal or mesenteric lymph nodes into the brains of these mice[2]. Both findings suggest a direct communication between the brain and the gut that is mediated by immune cells. Although it is clear that immune cells can traffic between the CNS and the periphery, the homing molecules that target these cells to the gut require further characterization. Through scRNA-seq, we were able

to identify the population of migrating macrophages that are specific to the brain and the gut. This migrating population of macrophages was enriched for the proposed αSyn-receptor protein, LRP1[27]. This protein has recently been shown to be responsible for αSyn uptake and transport, and could likely be the protein responsible for αSyn transport in these cells. LRP1 was found to mostly regulate the uptake of monomeric or oligomeric forms of αSyn, and to a much lesser extent aggregated αSyn in the form of pre-formed fibrils. This data indicates that mostly unaggregated forms of αSyn are taken up by migrating macrophages, and these subsequently aggregate in the ileum. Specific uptake of αSyn via a receptor is further supported by the lack of GFP trafficking from the brain to the ileum. As LRP1 is also responsible for apolipoprotein E (APOE) and amyloid-β (Aβ)[27], it would be important to test if the LRP1⁺ macrophages also play a role in other neurological disease such as Alzheimer's Disease.

## Methods

### Inclusion and ethics statement

The research described here adhere with all relevant ethical regulations. All experiments involving mice were approved by the local authorities at the Regierung von Unterfranken, Würzburg, Germany under experiment numbers: 55.2.2-2532-2-631-101, 55.2.2-2532-1455-34, and 55.2-2-2532-2-992. Patient autopsies were obtained according to the guidelines of the Local Ethics Committee of the University Hospital of Marburg.

### Mice

Mice were kept at the animal facility of the Center for Experimental Molecular Medicine, University of Würzburg, under barrier conditions and at a constant cycle of 12 h in the light and 12 h in the dark. Colonies

were maintained at 20–24° Celsius and 40–60% humidity, with free access to food and water. Ten-to-12-week-old male C57Bl/6J or SNCAKO (JAX stock#016123)[45] mice were obtained from Charles River Laboratories (Sulzfeld, Germany) and stereotactically injected unilaterally into the right SN with 2 µL of either empty AAV1/2 (EV), an AAV1/2 encoding the human mutated A53T-αSyn, or an AAV1/2 encoding GFP (GeneDetect) at a concentration of $5.1 \times 10^{12}$ genomic particles/mL using the coordinates: −3.1 mm anteroposterior (AP), −1.4 mm mediolateral (ML), and 4.4 mm dorsoventral (DV)[12]. Both hαSyn and GFP were expressed under the CMV promoter. For the transgenic A30P/A53T PD mouse model, male and female heterozygous hm2a-Syn-39 were obtained from Jackson Laboratory (Jax stock# 008239)[46] and 16–17-month-old animals used for the experiments. Homozygous male C57Bl/6 mito-Dendra2 mice were obtained from Jackson Laboratories (Jax stock# 018397)[47]. C57Bl/6J CD11c.DOG[18] mice were kindly provided by G. Hämmerling, interbred with albino C57Bl/6J mice and maintained in the Center for Experimental Molecular Medicine (ZEMM) animal facility at Würzburg University. Both the Dendra2 and the CD11c.DOG mice were male 10–12 weeks old at the time of stereotactic injection. The local authorities at the Regierung von Unterfranken, Würzburg, Germany approved all animal experiments. The Dendra2 experiments were performed under experiment number 55.2.2-2532-2-631-101 and the CD11c.DOG experiments were performed under experiment number 55.2.2-2532-1455-34. All other experiments were performed under experiment number 55.2-2-2532-2-992.

## Immunohistochemistry

Mice were sacrificed at either 1 week, 5 weeks, or 10 weeks following stereotactic surgery or at 16–17 months of age in the case of A30P/A53T transgenic animals. After transcardial perfusion with ice cold 0.1 M phosphate-buffered Saline (PBS)/heparin (Ratiopharm #X34331), mouse brains, spleens, cervical lymph nodes, thoracic section of the vagus nerve, distal ilea and distal colons were isolated and immediately embedded in TissueTek and snap frozen. Ten micrometer thick sections were obtained from the various organs and sections were stored at −20 °C. Following fixation with 4% Paraformaldehyde (Sigma #8.18715.1000) and a block with 2% Bovine Serum Albumin (BSA) (Sigma #A4503) and 10% Normal Goat Serum (NGS) (Dako #X0907), sections were incubated overnight with chicken anti-TH (1:500, Abcam #ab76442), rabbit anti-αSyn (1:10,000 brain and 1:5000 other organs, Sigma #S3062), rabbit anti-GFP (1:500, Invitrogen #A11122), rat anti-HA (1:100, Roche, #11867423001), rabbit anti-phospho αSyn S129 (1:100, Abcam, #ab59264), hamster anti-CD11c Alexa Fluor 488 (1:300, Invitrogen, #53-0114-82), rat anti-CD11b (1:100, Serotec, #MCA74G), goat anti-CHAT (1:300, Millipore, #Ab144P), rabbit anti-P2RY12 (1:200, Invitrogen, #PA5-77671), rabbit anti-LRP1 (1:100, Abcam, #Ab92544), rat ant-CD8 (1:500, Serotec, #MCA609G), rat anti-F4/80 (1:300, Serotec, #MCAP497), or rat anti-CD4 (1:1000, Serotec, MCA1767) in 2% NGS/1%BSA. Following three washes with 1× PBS, sections were then incubated for 1 h with the appropriate secondary antibodies diluted 1:300 in 2%NGS/1%BSA: goat anti-rabbit Cy3 (Abcam, #ab150175), goat anti-chicken AF647 (Abcam #ab150175), goat anti-rat AF647 (Abcam, #ab15016), donkey anti-goat (Dianova, #705-165-147). Sections were then washed and mounted with Fluoromount-G with DAPI (Invitrogen #00-495952). To measure proteinase K-resistant aggregations, sections were pre-treated with PK (20 µg/mL, Sigma #P2308) for 10 min at 37 °C prior to the blocking step. Complete digestion was verified by lack of TH signal.

αSyn, pαSyn, and PK resistant-containing villi were counted on two to three sections per animal, averaged and were quantified as per mm2 of ileal tissue. As αSyn was more diffuse in the SN, the median fluorescent intensity (MFI) of PK-resistant αSyn was quantified using ImageJ on a minimum of three sections per animal. Due to the lack of TH signal following PK digestion, the SNpc borders on the digested slices were defined with the selection tool in Fiji (ImageJ v. 2.3.0) by the

TH signal from undigested slice of the same region. The mean gray value was measured within the selection and background was subtracted by measuring the mean gray value on an area without signal. Values are depicted as relative to control EV animals that were stained and imaged on the same day.

## Immunohistochemistry on patient autopsies

Formalin-fixed and paraffin-embedded archived human brain tissue from the SN pars compacta were consecutively collected from neuropathologically characterized PD cases (n = 10) and controls (n = 4) matched for age, sex, and interval from death to tissue fixation were collected from the Department of Neuropathology Marburg with written informed consent from the respective responsible dependent and according to the guidelines of the Local Human Ethics Committee of the University Hospital of Marburg. The mean (±standard deviation) age was 74.3 ± 7.7 years in the control group and 73.0 ± 12.4 years in the PD group. The sex was obtained from the autopsy report and the male/female ratio in the control group was 3/1, in the PD group 7/3.

Double immunohistochemistry was performed on 3 µm thick sections with rabbit anti-CD11c (1:1000; Abcam EP1347Y; ab52632) combined with mouse alpha-synuclein (1:5000; Roboscreen; Mab 5G4) on the Leica Bond III platform. For the red staining of CD11c, the Bond Polymer Refine Red Detection Kit (DS9390) was used. For the green staining of alpha-synuclein, the Bond Polymer Refine detection Kit (DS9800) and Histogreen were used. CD11c+ cells with a visible nucleus within both SN of midbrains were counted using ×40 magnification and the amount of double labeled CD11c+αSyn+ cells of all CD11c+ cells per area was calculated.

## Gut transit test

A 6% solution of Carmine Red (Sigma, #C1022) was freshly prepared in 0.5% methylcellulose (Sigma, #M0512). Two hundred microliters of this solution was then administered to the mice using a 21 gauge round tip feeding needle. Mice were immediately single housed and stools were continuously monitored for Carmine Red. Total gut transit time is recorded as the interval between administration of Carmine Red and the first observance of it in the stool.

## Fecal pellet output

Mice were placed in an empty cage and fecal pellets were collected and weighed in a pre-weighed tube after 1 h. Pellets were dried at 65 °C overnight and tubes were re-weighed to assess the dry fecal pellet weight. The difference between the pre-dried weight and the post-dried weight was calculated to assess the water content percentage.

## Depletion of CD11c+ cells

Ten-to-12-week-old wildtype and transgenic B6a.CD11c.DOG mice were stereotactically injected with an AAV1/2 encoding the human mutated A53T-αSyn as described above. Starting 1 week after injection, mice were injected with 20 ng/g Diphtheriatoxin (Sigma, #D0564) in 200 µL PBS every second day for 2 weeks. Following the 2 weeks of treatment mice were sacrificed and organs harvested for further analysis.

## Western blot

Following isolation of WBCs from 200 µL of whole blood (described below). Samples were resuspended in 50 µL of Ripa Buffer (25 mM Tris pH 8.0, 10 mM Hepes, 150 mM NaCl, 145 mM KCl, 5 mM MgCl$_2$, 0.1% SDS, 1% NP-40, 10% Glycerol, 2 mM EDTA), sonicated and stored at −20 °C.

Ileal and SN sections were flash frozen in liquid nitrogen, weighed, and resuspended in RIPA buffer. Ileal samples were disrupted using the stainless-steel beads in the TissueLyser LT (Qiagen), then centrifuged at 15,781 G for 5 min. Supernatant was collected and stored at −20 °C.

For all samples, protein concentration was measured using the Lowry reagent and samples were mixed with the appropriate amount

of 5× Loading buffer (250 mM Tris/HCl pH6.8, 25% Beta-mercaptoethanol, 20% SDS, 50% Glycerin, 0.20% Bromphenolblue). Ten micrograms of protein were loaded onto a 12% SDS-Gel and after running, protein was transferred onto a pre-wet Nitrocellulose membrane. Membranes were blocked with 5% milk and incubated overnight at 4 °C with rabbit anti-phospho αSyn S129 (1:1000, Abcam, #ab59264), rabbit anti-αSyn antibody (1:200, Abcam #212184), rabbit anti-P2RY12 (1:500, Invitrogen, #PA5-77671), rabbit anti-GFP (1:500, Invitrogen #A11122), or mouse anti-GAPDH (1:15,000, Calbiochem, #CB1001). After washing, membranes were incubated for 1 h at RT with the appropriate peroxidase-conjugated secondary antibodies diluted 1:5000: anti-rat IgG (Jackson Immuno research, #712-035-150), anti-rabbit IgG (Jackson Immuno Research, #711-035-152), or anti-mouse IgG (Jackson Immuno Research, 715-035-150). Blots were then developed using Western Lightning Plus (Perkin Elmer, NEL103E001EA).

### Isolation of lamina propria leukocytes

Ten-centimeter-long sections of the ileum were isolated as described above and placed on a PBS soaked paper towel. Feces, fat, and peyers patches were removed and ilea were placed in ice cold RPMI (Sigma #R8758) supplemented with 10% Fetal Bovine Serum (Sigma #F7524) (complete RPMI). The epithelial layer was removed by incubating the tissues at 37 °C for 30 min in complete RPMI supplemented with 2 mM EDTA (Millipore #324506). Ilea were then washed three times with PBS and incubated for 15 min at 37 °C in Accutase (Sigma #A6964). Digested samples were then passed over a 70 μm cell strainer, centrifuged, and resuspended in 40% Percoll (GE Healthcare #17-0891-02). Samples were centrifuged at 800 G for 25 min at RT and the pellet containing the leukocytes was collected for further experiments.

### Flow cytometry

After blocking with CD16/32 (1:200, Biolegend #101302) for 10 min, cells were stained with Fixable Violet Dead Cells Stain kit (1:1000, Invitrogen, L34955) together with rat anti-I-A/I-E Cy7(1:100, Biolegend, 107630), rat anti-CD86 PE (1:50, Invitrogen, 12-0862-82), hamster anti-CD103 AF488 (1:100, Biolegend, 121408), hamster anti-CD11b PerCP/Cy55.5 (1:50, Biolegend, 101228), hamster anti-CD11c APC (1:100, Biolegend, 117310), rat anti-CD45 APC (1:200, Biolegend #103111), rat anti-CD4 PE (1:100, BD Biosciences #553049), rat anti-CD8 PerCp/Cy5.5 (1:100, Biolegend #100733) for 30 min at 4 °C. After a wash, cells were either analyzed immediately or fixed and permeabilized using the FoxP3 staining kit. Cells were then incubated with a monoclonal rabbit anti-αSyn antibody (1:200, Abcam #212184) for 30 min followed by incubation with an goat anti-rabbit Cy5 antibody (1:300, Jackson Biozol #711-175-152). Flow cytometry analysis was performed using Flowjo v. 10.8.1 (BD Biosciences).

### Isolation of immune cells

Brains were isolated as described above and placed in ice-cold PBS. After finely mincing the brains, Accutase was added and the brains were incubated at 37 °C for 30 min while shaking. Following digestion, brains were pressed through a 70 μm cell strainer and resuspended in 40% Percoll. Samples were centrifuged at 650 G for 25 min and the pellets were isolated and used for further experiments.

Spleens and LN were pressed through a 70 μm filter and incubated with red blood cell (RBC) lysis buffer (0.15 M NH4Cl, 10 mM KHCO3, 0.1 mM Na2EDTA) for 3 min. Cells were centrifuged down and used for further experiments.

Peripheral blood was incubated in RBC lysis buffer for 5 min prior to centrifugation.

### Single-cell sequencing and analysis

Five animals per group were sacrificed and single cell suspensions from the brain, ileum, and spleen cells were made. An equal number of cells from similar organs were pooled and 20,000 CD4[+], CD8[+], and CD11c[+] cells/organ/condition were FACS-sorted using a FACSAria III (BD). Single cell sequencing libraries were generated from the pooled biological replicates with the Chromium Single Cell 3′ v.3 assay (10× Genomics) according to the company's protocol. Libraries were sequenced with the NovaSeq 6000 platform (S1flow cell, 100 cycles; Illumina) in paired-end mode to reach an average depth of 125,000 reads and 2300 genes per single cell. Data were demultiplexed using the CellRanger software v. 6.1.2. The reads were aligned to a custom mm10 reference genome which included the complete hαSyn gene using the STAR aligner.

Data analysis was performed using the Seurat v. 4.1.1 R package[48]. Low quality cells with more than 20% mitochondrial genes and <200 or >7500 genes were removed. EV and hαSyn data for each organ was merged together and the data was then log-normalized. The FindVariableFeatures function was used to select variable genes. The ScaleData function was then used to scale the data prior to Principal Component Analysis, UMAP dimensional reduction, and clustering using 15 principal components and a resolution of 0.5. The cluster markers were found using the FindAllMarkers function and T cells were removed based on cluster markers. Brain, Spleen, and Ileum CD11c[+] cells were then integrated together using the IntegrateData function. CD11c[+] clusters were then formed using the FindClusters function and a resolution of 0.6. FindAllMarkers was again performed and the clusters identified using the Mygeneset tool in the Immunological Genome Project[19]. Gene Ontology analysis and subsequent plots were created using cluster profiler v. 4.8.3 on the respective marker gene sets[49].

### Cytokine analysis

Total protein was isolated from SN and Ileal samples as described above. Protein concentrations were measured and samples were diluted with RIPA Buffer to 5 μg/μL for the SN and 8 μg/μL for the ileum. In total 11 μL of each sample was loaded onto a CodePlex Chip accordingly to manufacturer's instructions (PhenomeX, #CODEPLEX-2L04-2), followed by loading the CodePlex Chip into the Isolight reader (IsoLight software version 1.10.0, PhenomeX). Cytokine concentrations were then measured using the corresponding CodePlex Secretome Adaptive Immune – Mouse Kit (PhenomeX). Raw data was analyzed by the automated IsoSpeak software.

### Photoconversion of cells in the SN

Ten-to-12-week-old Dendra2 mice were stereotactically injected with AAV as described above. Immediately after injection, a mono fiber-optic cannula was implanted directly above the SN (AP:3.1, ML:1.4, DV: 4.1) and secured. Starting one week following the surgery, the SN of the mice were exposed weekly to 40 mW of light at a wavelength of 473 nm for 30 s via the fiber-optic cannula. Mice were sacrificed one day following the last photoconversion session, and the blood, cervical lymph nodes, spleen, ileum, and brain were harvested.

### Statistical analyses

To determine the normality of the data, a Q-Q plot was analyzed in GraphPad v. 9.4.0. For normally distributed data, parametric methods were utilized. For non-normal distributed data, we utilized non-parametric methods. For the TH[+] and Nissl[+] stereological investigations, one-way ANOVA with Tukey's multiple comparison post hoc test was utilized. Two-way ANOVA with Bonferroni's post hoc test was utilized for αSyn[+] and pαSyn[+] villi assessment, Fecal Pellet Output, Fecal Water Content, CD86 expression, and CD11c[+] photoconverted cells. Unpaired one-tailed Student's $t$ test was utilized to analyze αSyn MFI, αSyn[+]CD11c[+] cell numbers, cytokine expression, and percent photoconverted cells in the ileum. $p$ Values of <0.05 were considered significant. GO Term Enrichment was determined using a one-sided hypergeometric test and the Benjamini–Hochberg correction for multiple comparisons.

## Reporting summary

Further information on research design is available in the Nature Portfolio Reporting Summary linked to this article.

## Data availability

The sequencing datasets generated and analyzed during the current study are available at GEO (http://www.ncbi.nlm.nih.gov/geo) under the accession number GSE232840. The experiment data that support the findings of this study are available upon request. Source data are provided with this paper.

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

## Acknowledgements

This project was supported by the Alexander von Humboldt-Stiftung (R.L.M.), by the Bavarian State Ministry of Economic Affairs, Regional Development and Energy within the project: Single-cell analysis in personalized medicine at the Helmholtz-Institute for RNA-based Infection Research implemented in the Single-Cell Center Würzburg (C.W.I.), by the Deutsche Forschungsgemeinschaft (DFG, German Research Foundation) Project-ID 424778381-TRR 295 (A01, A06 to C.W.I. and J. Volkmann), Project ID 324392634 (B09 to A.B.), and by the Interdisciplinary Center for Clinical Research (IZKF) at the University of Würzburg (A-303, A-421, N-362, S-506 to C.W.I.). Moreover, C.W.I. is supported by the VERUM Foundation. J. Volkmann has received funding from the European Union's Horizon 2020 research and innovation program under the EJP RD COFUND-EJP No. 825575 (EurDyscover). The authors are grateful to Veronika Senger, Louisa Friess, Jasmin Lang, Selin Asci, and Heike Menzel for their excellent technical assistance.

## Author contributions

R.L.M. and C.W.I. conceived the ideas and designed the experiments. R.L.M., A.M., J.G., U.K., F.I., J.P.M., A.P., S.T., J.W., J.H., and A.K. performed the experiments and analyzed the data. R.L.M., A.P., J. Vogel, A.B., A.E.S., J.K., J.B., J. Volkmann, and C.W.I. interpreted the data. All authors reviewed, edited, and approved the manuscript.

## Funding

## Competing interests

The authors declare no competing interests.
