## [Peer Review File · Nature Communications]

Brain-to-gut trafficking of alpha-Synuclein by CD11c+ cells in a mouse model of Parkinson's DiseaseREVIEWER COMMENTS

Reviewer #1 (Remarks to the Author):

Summary

In the manuscript by McFleder and colleagues, the authors present the novel observation that exogenous alpha synuclein introduced into the brain can be trafficked to the gut by CD11c+ immune cells. The significance of this trafficking on disease pathology and the identification of the immune populations that transport alpha synuclein are key questions. Although the authors present data to indicate intestinal dysfunction (increased fecal output), the data does not sufficiently indicate intestinal pathology, as other explanations for the data are possible. Therefore, it is not clear if the trafficking of exogenous alpha synuclein is associated with any dysfunction in the intestine. Further experiments would help to confirm intestinal dysfunction (suggestions below), if present. The authors examine the CD11c+ populations through flow cytometry, immunohistochemistry (IHC), and single-cell RNA sequencing (scRNA-seq). These data provide good characterization of the types of CD11c+ cells in the brain and ileum. However, the data does not conclusively identify the type of myeloid population or populations that are transporting the exogenous alpha synuclein (HA-tagged). Further examination of HA+ cells through immunohistochemistry may be able to resolve the population subtypes of those cells. Finally, the authors demonstrate that CD11c+ cells can migrate from the brain to the gut using transgenic mice with photoconvertible fluorescent protein expression. It is surprising not to see trafficking from the brain to the brain-draining lymph nodes or spleen, as these are important immune cell hubs for interactions of myeloid cell populations with T cells. However, in the field, there are many unanswered questions about immune cell trafficking to and from the brain. Overall, the data presented provide some new areas to explore in the trafficking of immune cells and alpha synuclein transport. The data presented were obtained with appropriate technique and statistical analysis. In the introduction and discussion, the manuscript references previous literature appropriately. Further experiments, as suggested below, would help to address the key questions of significance and the identity of the populations involved.

Suggested Improvements

Figure 1/Extended Data Figure 1:

1D: Quantification of HA+ villi in the ileum would inform the reader on the scale of the

transport of exogenous human alpha synuclein from the brain to the intestine.

1h & 1l: Additional data beyond fecal output and water content is needed to determine if there is a pathological influence on gut function in the AAV model presented. Fecal output and water content are not sufficient evidence of intestinal pathology, as other causes for the increase in fecal output could be hyperactivity, anxiety, and/or response to novel environment (Wang L, NeuroReport, 2008; doi: 10.1097/WNR.0b013e3282ffda5e).

Additionally, most PD models (e.g., A53T transgenic mice) have shown reduced fecal output, not increased as indicated in the data. Few have shown an increase in fecal output in PD models and, where observed, it was attributed to novel environment (Wang L, NeuroReport, 2008). Possible measurements that would support the authors' assertion of perturbation in gastrointestinal function include, total intestinal transit time, as measured by Carmine Red or other dye, geometric center measurements by dye/reporter, bead expulsion assay, or colonic motor complex measurements. One or more results from the assays above indicating dysfunction would provide stronger evidence. It is not clear with the current data that there is a pathophysiological impact on the intestine as claimed or that aSyn protein accumulation is involved.

Figure 2:

2C & 2E: It is not clear if the trafficking of exogenous alpha synuclein is associated with any intestinal dysfunction. Hallett et al. (Hallett PJ, Neurobiol Dis., 2012) demonstrated that endogenous and insoluble (proteinase K-resistant) mouse alpha synuclein can form in the intestines of wild type mice. This is also observed in the empty vector (EV) control mice in this report, where small amounts of PK-resistant and phospho-alpha synuclein can be measured (Figures 2C and 2E). Although the data demonstrate that insoluble alpha synuclein is accumulated in the brain, with sustained increase over control mice from week 1 through 10 (Figure 2A & 2B), the data in the intestines is not as strong (Figure 2C).

Increased PK-resistant alpha synuclein is observed early (1 week) but not at later timepoints (5 and 10 weeks). This does not correspond to the observed fecal output phenotype observed, which occurs at 10 weeks, but not earlier time points. Additional, tests of intestinal dysfunction(as listed above) may find gut dysfunction earlier, if present.

Figure 3/Extended Data Figure 2:

The mouse intestine contains multiple myeloid populations, including monocytes, macrophages, and dendritic cells. CD11c, in general, is a marker of dendritic populations but

may be upregulated during inflammation and appear on monocytes and macrophages. In the small intestine, there are populations of CD103-positive and -negative dendritic cells (Bogunovic M et al., *Immunol Res*, 2012; Joeris T et al., *Mucosal Immunology* 2017; Andersen DA, *Nat Rev Immunol*, 2021). Without additional markers, it is not clear if the HA+ or aSyn+ CD11c+CD103- populations are macrophages or dendritic cells. It is more accurate to refer to the population as CD11c+ myeloid cells or mononuclear phagocytes (MNP). Additional characterization of the HA+CD11c+ population through marker staining may help clarify the specific subpopulation of immune cells.

Figure 4:

4D-F: With only single and pooled samples for each organ and treatment group in the single-cell RNA-seq data, it is difficult to draw conclusions about changes in immune cell population numbers or function in the brain or ileum. The changes in cell numbers in Figure 4D may represent variation in mice or sample preparation; additional data would help to confirm the scRNA-seq observations (e.g., flow cytometry or IHC). Changes in gene transcripts do not always correlate with changes in protein expression. In order to support the assertion that pro-inflammatory CD11c+ populations are increasing in the brain and ileum (Figure 4F), it would be useful to have some other measurements of inflammation, such as secreted cytokine quantification (e.g., Tnf, Ccl3, or Ccl4).

Figure 5/Extended Data Figure 3 and 4:

With the scRNA-seq data, there is an opportunity to probe the identity of the migrating population from the brain to the gut. Additional staining (e.g., IHC) for population/cluster-specific markers (from the scRNA-seq data) may help clarify the nature of the HA+ population in the ileum.

The co-expression of Lrp1 in HA+ cells is an interesting observation. Can the Lrp1+HA+ cells be quantified (Figure 5E)? It is difficult to assess the extent of enrichment of Lrp1+ in HA+ cells from a single representative image.

Figure 6.

Figure 6A, 6D & 6E: In the representative plots, what population is represented? Is it total live cells in 6A and 6D, and Live/Dendra2_Red+ in 6E? Please clarify the pre-gating in the figure legend.

Reviewer #2 (Remarks to the Author):

NCOMMS-23-20193

Brain-to-gut trafficking of alpha-Synuclein by CD11c+ cells in a mouse model of Parkinson's Disease

Inflammation in the brain and gut is a critical component of several neurological diseases, such as Parkinson's disease (PD). One trigger of the immune system in PD is aggregation of the pre-synaptic protein, alpha-synuclein (alphaSyn). Understanding the mechanism of propagation of alphaSyn aggregates is essential to developing disease-modifying therapeutics. Using a brain-first mouse model of PD, the authors demonstrate alphaSyn trafficking from the brain to the ileum of mice. Immunohistochemistry (IHC) revealed that ileal alphaSyn aggregated prior to the brain and was contained within CD11c+ cells. Using single-cell RNA sequencing (scRNA-seq), the authors demonstrate that ileal CD11c+ cells are microglia-like, and the same subtype of cells is activated in the brain and ileum of PD mice. Moreover, by utilizing mice expressing the photo-convertible protein, Dendra2, they show that CD11c+ cells traffic from the brain to the ileum. Together these data provide a novel mechanism of alphaSyn trafficking between the brain and gut.

The data presented suggest a connection in PD mice between the brain and the ileum, mediated by CD11c+ macrophages. It would be instructive to show that depletion of CD11c+ macrophages eliminate the appearance of pathologic alphaSyn in the ileum. This experiment seems essential to the authors hypothesis.

Reviewer #3 (Remarks to the Author):

The study by McFleder and coauthors is an interesting investigation into the dynamics of myeloid cells and their relevance to alpha-synuclein (α -syn) pathology in a mouse model, with important implications for translational immunology as well as human neurodegenerative disease. The authors utilize a broad range of modern techniques which enable the elegant investigation of the research hypothesis. However, some of the key conclusions are not supported by the current data and need either to be demonstrated with additional analysis/data, or to be adjusted.

Basically, the study proposes three key mechanisms: 1) Localized, vector-mediated aSyn overexpression in the brain results in aSyn pathology in the gut independently from vagal transmission 2) CD11c+ cells rapidly circulate b/w brain and ileum in physiological and pathological conditions 3) These CD11c+ cells carry aSyn protein from the brain to the ileum. The first two mechanisms are somewhat novel, very-well reasoned and backed by the data. However, the third mechanism, which has the highest novelty and potential therapeutic implications, is based on several assumptions that are not fully backed by the data presented.

The first two mechanisms already render the study worth publishing, but the third mechanism needs to be demonstrated better by the data or adjusted to avoid overstatement.

Please find below major and minor points for consideration:

Major point:

1) The main problem is with the concept that the aggregated aSyn in the ileum is produced by the AAV-transduced neurons in the brain, -> released by them -> taken up by the CD11c+ cells -> and transported to the ileum. While this seems highly likely, no data is presented to exclude the possibility that the aggregated aSyn in the ileum is produced by the ileum-residing CD11c+ themselves, because they were also transduced in the brain at the time of injection. If this is the case, this will drastically reduce the relevance to the natural human disease.

There are several relatively easy ways to disprove this:

- Explicitly demonstrate that the mouse CD11c+ cells are not transduced by AAV1/2, either with own data or by citing robust literature studies.

- Demonstrate with RT-qPCR/FISH that the vector and its transcripts are not present in the ileum tissue and best also in the circulating WBCs, for example with a forward primer for human aSyn and a reverse primer for HA-tag/BGH/WPRE. Optimally, this will be shown shortly after treatment, as the authors show that the CD11c+ cells move very quickly between tissues.

- Re-analyze the single-cell data (see below why) to demonstrate that the CD11c+ cells really do not express the transgene themselves

The GFP AAV control provides some support here, however has several problems:

- It would require to show a quantification of GFP-positive cells/villi in Ileum at 1, 5 and 10 weeks after treatment (even if all values are 0), like in Figure 1 E, or even better like in Fig. 2 B with MFI quantification; instead of the single illustrative IHC picture showed in Figure 1 D, third panel. In fact, the low magnification of Fig 1 D does not allow the reader to appreciate if there are indeed no single GFP+ cells. Actually, even if GFP-positive cells are found in the ileum, this will only disprove the specificity of the process to aSyn; it won't necessarily mean that the cells were transduced by the vector. To disprove this, the aforementioned data/analysis/literature is crucial.

- The authors show that WBC carry aSyn (Ext. Data Fig. 2D & E). It would be interesting to see/exclude if they also carry GFP with the same method. If the WBCs carry GFP, but are not found in the ileum, this will have some interesting implications.

Similarly, the single-cell data can offer some support here. However, its interpretation has one major flaw: The authors state that the lack of aSyn mRNA (Snca gene) in the single-cell data demonstrates that aSyn in CD11c+ cells is taken up and not expressed there (Discussion L256-262). However, they utilized the 10X Genomics 3' GEX kit, which detects almost only the last 400-500 bp of a transcript (3'). Thus, the transgene as described in Kolprich et al. Mol Degener 2010 5:43; will not be assigned to the Snca gene, as only the HA/WPRE/polyA/BGH portion 'behind' human aSyn will be visible to the aligner. I would recommend aligning to a custom mm10 reference including the exact sequence of the transgene transcript with the 5' and 3' sequences and visualizing the alignments on a genomic browser. Also, I think the transgene is very stable for a long time in transduced cells, but what if the cells do not express it at 5 weeks after treatment (time point of the single-cell data) just because it was already degraded/inactivated? Could the authors comment on this?

Finally, even if the authors cannot unequivocally solve the major concern, the hypothesis remains novel, highly likely and worthy; but then at least the conclusions need to be adjusted. For example, L247 "Through the use of an HA-tag, our study definitively shows that α Syn molecules traffic out of the brain and to the gut. " is an overstatement if the authors do not show data or literature that excludes the possibility that the CD11c+ cells

were expressing the transgene themselves, possibly even only after leaving the brain.

Minor points:

2) Fig 2B: Looking at the distribution of the data in 2B, aSyn can be considered aggregating ALSO already at 1 and 5 weeks, what are the p-values here? Related to this, line 263-264 from the discussion “Despite being a brain-first model of PD, the h α Syn animals demonstrated α Syn aggregations in the

264 ileum prior to significant accumulations in the brain.” is an exaggeration, since there is obviously an increase in MFI of PK-resistant aSyn MFI in SNpc at 1 week post treatment, even if the p-value is not below 0.05. To ‘prove’ the lack of difference, the authors would have to report the post-hoc statistical power β for detecting a difference of at least the magnitude which is seen at 10 weeks.

3) Results L118-L220 “This suggests that expression of the mutated human α Syn leads to aggregation of endogenous α Syn, likely explaining why mostly endogenous α Syn propagates to the ileum in this model.”: What is the justification that it shows exactly that? It could be e.g. that the inflammation caused by the model results in the expression of the endogenous aSyn.

4) Related to the major concern, I suggest using more concise wording: for the presence of aSyn protein from the brain in the ileum, use “presence”, not expression, which can be understood here as gene expression.

5) L136: “...tended to co-localize...” is misleading, as it is obvious in the IHC that there are many other aSyn-staining cells that are not CD11c (green) and vice versa. Maybe re-phrase?

7) The legend of Fig. 3 D specifies red and green in the IHC staining, are these the correct colors? It is hard to appreciate in the figure.

8) Seems like there are two very distinct groups of PD patients in Fig. 3 E, can the authors comment on this?

9) Fig. 4 D – The most profound change of cell type proportions is in the spleen, which

seems contra-intuitive to the main hypothesis, why could that be?

10) A part of the last sentence of Fig. 4 legend is missing.

11) Fig. 6D: The authors state that there are no Dendra-Red-positive cells in the LNs and Spleen; here the quantification (percent of Dendra-Red-positive cells) in all 8 animals is more appropriate than one representative plot - obviously these are the 8 animals that are shown in Fig 6. C, of which 4 animals have very low number of photoconverted cells also in the ileum and brain. Do the 3 animals with high frequency (> 0.1 %) of photoconverted cells in the brain also show 0 % converted cells in LNs and spleen?

12) GO analysis – Since these cells were all gated on their high(er) expression of CD11c, which an important part of leucocyte migration, and often co-regulated with other leucocyte migration molecules, it comes to no surprise that comparing gene sets from this cells to the whole mouse genome returns enrichment of leucocyte migration terms. It would be good if the authors show as extended data if the top 10 GO terms of the other populations markers do not include leucocyte migration terms to prove their point that this is a characteristic of the TRM1/TRM2/M1 cells.

Response to Reviewers

Reviewer #1 (Remarks to the Author):

Summary

In the manuscript by McFleder and colleagues, the authors present the novel observation that exogenous alpha synuclein introduced into the brain can be trafficked to the gut by CD11c+ immune cells. The significance of this trafficking on disease pathology and the identification of the immune populations that transport alpha synuclein are key questions. Although the authors present data to indicate intestinal dysfunction (increased fecal output), the data does not sufficiently indicate intestinal pathology, as other explanations for the data are possible. Therefore, it is not clear if the trafficking of exogenous alpha synuclein is associated with any dysfunction in the intestine. Further experiments would help to confirm intestinal dysfunction (suggestions below), if present. The authors examine the CD11c+ populations through flow cytometry, immunohistochemistry (IHC), and single-cell RNA sequencing (scRNA-seq). These data provide good characterization of the types of CD11c+ cells in the brain and ileum. However, the data does not conclusively identify the type of myeloid population or populations that are transporting the exogenous alpha synuclein (HA-tagged). Further examination of HA+ cells through immunohistochemistry may be able to resolve the population subtypes of those cells. Finally, the authors demonstrate that CD11c+ cells can migrate from the brain to the gut using transgenic mice with photoconvertible fluorescent protein expression. It is surprising not to see trafficking from the brain to the brain-draining lymph nodes or spleen, as these are important immune cell hubs for interactions of myeloid cell populations with T cells. However, in the field, there are many unanswered questions about immune cell trafficking to and from the brain. Overall, the data presented provide some new areas to explore in the trafficking of immune cells and alpha synuclein transport. The data presented were obtained with appropriate technique and statistical analysis. In the introduction and discussion, the manuscript references previous literature appropriately. Further experiments, as suggested below, would help to address the key questions of significance and the identity of the populations involved.

Suggested Improvements

Figure 1/Extended Data Figure 1:

1D: Quantification of HA+ villi in the ileum would inform the reader on the scale of the transport of exogenous human alpha synuclein from the brain to the intestine.

We appreciate the valuable suggestion of reviewer 1. To better inform the reader regarding the amount of HA and human α -synuclein transport, we have quantified the HA⁺ villi in the h α Syn PD mice compared to EV control and show the numbers in the revised Figure 1 (1F). In addition, we describe the transport in the results in line 96.

1h & 1l: Additional data beyond fecal output and water content is needed to determine if there is a pathological influence on gut function in the AAV model presented. Fecal output and water content are not sufficient evidence of intestinal pathology, as other causes for the increase in fecal output could be hyperactivity, anxiety, and/or response to novel environment (Wang L, NeuroReport, 2008; doi: 10.1097/WNR.0b013e3282ffda5e). Additionally, most PD models (e.g., A53T transgenic mice) have shown reduced fecal output, not increased as indicated in the data. Few have shown an increase in fecal output in PD models and, where observed, it was attributed to novel environment (Wang L, NeuroReport, 2008). Possible measurements that would support the authors' assertion of perturbation in gastrointestinal function include, total intestinal transit time, as measured by Carmine Red or other dye, geometric center measurements by dye/reporter, bead expulsion assay, or colonic

motor complex measurements. One or more results from the assays above indicating dysfunction would provide stronger evidence. It is not clear with the current data that there is a pathophysiological impact on the intestine as claimed or that aSyn protein accumulation is involved.

We thank this reviewer for pointing out this important issue. We agree that many gut-first and transgenic PD mouse models that induce altered α Syn expression in gut neurons exhibit reduced fecal output (Challis et al. 2020; Rota et al. 2019). However in other brain-first models, such as the MPTP and 6-OHDA model, fecal pellet output was increased similar to what we see in our brain-first model (McQuade et al. 2021). In response to the reviewer's concern, we also analyzed whole gut transit by utilizing a Carmine Red dye. Consistent with our previous results, we also detected decreased whole gut transit time in our PD animals compared to controls. As decreased gut transit time is also associated with gut inflammation (Yan Chen et al. 2021), this could serve as a further proof of the inflammatory processes occurring in the gut in our brain-first model of PD. Indeed the MitoPark PD mouse model where the mitochondrial transcription factor A is removed from dopaminergic neurons also exhibits decreased whole gut transit time (Ghaisas et al. 2019). This decreased gut transit time was due to increased peristalsis in the small intestines associated with inflammation, despite reduced peristalsis in the colon. Perhaps at later timepoints, after pathology has spread to the intestinal neurons our brain model will also exhibit signs of constipation. We have added this analysis to Figure 1J and updated the discussion with this information at line 295:

"The haSyn mice unexpectedly exhibited increased fecal pellet output and decreased whole gut transit. This is in contrast to other mouse models of PD that involve the enteric neurons and exhibit decreased fecal pellet output. This is likely a result of the inflammation occurring in the ileum at this time and could represent an early phase of disease before the onset of constipation resulting from the involvement of the enteric neurons. These results are similar to what was found in the MitoPark mouse model, which also exhibits intestinal inflammation and decreased whole gut transit time."

Figure 2:

2C & 2E: It is not clear if the trafficking of exogenous alpha synuclein is associated with any intestinal dysfunction. Hallett et al. (Hallett PJ, Neurobiol Dis., 2012) demonstrated that endogenous and insoluble (proteinase K-resistant) mouse alpha synuclein can form in the intestines of wild type mice. This is also observed in the empty vector (EV) control mice in this report, where small amounts of PK-resistant and phospho-alpha synuclein can be measured (Figures 2C and 2E). Although the data demonstrate that insoluble alpha synuclein is accumulated in the brain, with sustained increase over control mice from week 1 through 10 (Figure 2A & 2B), the data in the intestines is not as strong (Figure 2C). Increased PK-resistant alpha synuclein is observed early (1 week) but not at later timepoints (5 and 10 weeks). This does not correspond to the observed fecal output phenotype observed, which occurs at 10 weeks, but not earlier time points. Additional, tests of intestinal dysfunction (as listed above) may find gut dysfunction earlier, if present.

As described above, we analyzed gut transit time in our animals at 1, 5 and 10 weeks following AAV injection. Consistent with our previous results, we could detect significantly decreased gut transit time in PD animals by employing whole gut transit analysis via Carmine Red dye. By using this more sensitive assay of gastrointestinal dysfunction we could already detect significant changes at the 5 week timepoint. As 5 weeks is the timepoint when we see the most α Syn in the gut (Fig 1E), we believe this truly indicates that the trafficking of α Syn is associated with intestinal dysfunction. As indicated above, we have added this new analysis to the Figure 1J and described the results in line 105.

Figure 3/Extended Data Figure 2:

The mouse intestine contains multiple myeloid populations, including monocytes, macrophages, and dendritic cells. CD11c, in general, is a marker of dendritic populations but may be upregulated during inflammation and appear on monocytes and macrophages. In the small intestine, there are populations of CD103-positive and -negative dendritic cells (Bogunovic M et al., *Immunol Res*, 2012; Joeris T et al., *Mucosal Immunology* 2017; Andersen DA, *Nat Rev Immunol*, 2021). Without additional markers, it is not clear if the HA+ or aSyn+ CD11c+CD103- populations are macrophages or dendritic cells. It is more accurate to refer to the population as CD11c+ myeloid cells or mononuclear phagocytes (MNPs). Additional characterization of the HA+CD11c+ population through marker staining may help clarify the specific subpopulation of immune cells.

The reviewer raises an important point. We have performed additional stainings for the macrophage marker F4/80 and found that the majority (~80%) of the α Syn⁺ cells are indeed F4/80⁺. This extra staining in addition to the sequencing data demonstrating the migrating cells are macrophages, reaffirms our hypothesis that these cells are indeed macrophages. We have included the new staining in Extended Data Fig. 4H and modified the manuscript at line 246 with these new data as follows:

“Around 80% of the ileal h α Syn-containing cells were also positive for the macrophage-specific marker, F4/80 (Extended Data Fig. 4H).”

Figure 4:

4D-F: With only single and pooled samples for each organ and treatment group in the single-cell RNA-seq data, it is difficult to draw conclusions about changes in immune cell population numbers or function in the brain or ileum. The changes in cell numbers in Figure 4D may represent variation in mice or sample preparation; additional data would help to confirm the scRNA-seq observations (e.g., flow cytometry or IHC). Changes in gene transcripts do not always correlate with changes in protein expression. In order to support the assertion that pro-inflammatory CD11c⁺ populations are increasing in the brain and ileum (Figure 4F), it would be useful to have some other measurements of inflammation, such as secreted cytokine quantification (e.g., Tnf, Ccl3, or Ccl4).

In response to the reviewers concern we have performed cytokine analysis on bulk tissue lysates from the SN and the ileum of mice, 5 weeks after disease induction. Here we could find an elevation of CCL3 in the h α Syn ileal samples consistent with an increased proinflammatory CD11c⁺ population. The CCL3 in the bulk SN samples was unfortunately under the detection levels of our assay. We were however able to see an elevation of CXCL10 in the SN samples. As CXCL10 is a well-known cytokine associated with microglial and macrophage activation (Clarner et al. 2015), this demonstrates an increase in pro-inflammatory microglia in the brain as well. In our previous publication, we have also described an increase in activated microglia in this model (Karikari et al. 2022). Together with our new cytokine analysis, these data support our sequencing data that there is an increase in pro-inflammatory tissue resident CD11c⁺ macrophages in the SN and ileum of the h α Syn mouse model. This new data is added to Extended Data Fig. 4D & 4E and line 219 as follows:

“Cytokine analysis of bulk protein from the ileum likewise revealed an increased in CCL3 protein levels (Extended Data Fig. 4D). Although the CCL3 in the SN was below the detection limits, CXCL10 was significantly increased (Extended Data Fig. 4E). CXCL10 is classically associated with pro-inflammatory microglia and macrophages²⁴. Together these data suggest that h α Syn expression in the brain activates pro-inflammatory microglia-like populations in the brain and the ileum.”

Figure 5/Extended Data Figure 3 and 4:

With the scRNA-seq data, there is an opportunity to probe the identity of the migrating population from the brain to the gut. Additional staining (e.g., IHC) for population/cluster-specific markers (from the scRNA-seq data) may help clarify the nature of the HA⁺ population in the ileum.

The co-expression of Lrp1 in HA⁺ cells is an interesting observation. Can the Lrp1⁺HA⁺ cells be quantified (Figure 5E)? It is difficult to assess the extent of enrichment of Lrp1⁺ in HA⁺ cells from a single representative image.

As mentioned above we have probed the population further with the macrophage marker F4/80 and have demonstrated that nearly all of the α Syn⁺ profiles are contained within F4/80⁺ cells. We have also performed additional Lrp1/HA staining and quantification and demonstrated that at early timepoints around 40% of HA⁺ cells are Lrp1⁺. The Lrp1⁻ cells could represent HA that was released by the CD11c⁺ macrophages and taken up by other cells. In addition, depletion of CD11c dramatically reduced the number of α Syn⁺ cells (Fig. 3I). These additional experiments support the hypothesis that a substantial proportion of the cells trafficking α Syn are the CD11c⁺/F4-80⁺/Lrp1⁺ Macrophage 1 population. We have added a representative image in Extended Data Fig. 4H and the LRP1/HA quantification in Extended Data Fig. 4G. We have also added these results to the manuscript at line 242:

“By performing IHC for LRP1 in addition to HA, we were able to show that around 40% of ileal cells containing HA were also enriched for the α Syn-receptor, LRP1 (Fig. 5E & Extended Data Fig. 4G). Interestingly this was not true at the 10-week timepoint, likely reflecting a spread of the h α Syn to other cell types at later timepoints. Around 80% of the ileal h α Syn-containing cells were also positive for the macrophage-specific marker, F4/80 (Extended Data Fig. 4H).”

Figure 6.

Figure 6A, 6D & 6E: In the representative plots, what population is represented? Is it total live cells in 6A and 6D, and Live/Dendra2_Red⁺ in 6E? Please clarify the pre-gating in the figure legend.

We thank the reviewer for pointing out the need for this clarification. The parent population for 6A and 6D are total live green fluorescent cells and for 6E are live/Dendra2_Green/Dendra2_Red cells. We have added the gating strategy to Extended Figure 5B.

Reviewer #2 (Remarks to the Author):

NCOMMS-23-20193

Brain-to-gut trafficking of alpha-Synuclein by CD11c⁺ cells in a mouse model of Parkinson's Disease

Inflammation in the brain and gut is a critical component of several neurological diseases, such as Parkinson's disease (PD). One trigger of the immune system in PD is aggregation of the pre-synaptic protein, alpha-synuclein (alphaSyn). Understanding the mechanism of propagation of alphaSyn aggregates is essential to developing disease-modifying therapeutics. Using a brain-first mouse model of PD, the authors demonstrate alphaSyn trafficking from the brain to the ileum of mice. Immunohistochemistry (IHC) revealed that ileal alphaSyn aggregated prior to the brain and was contained within CD11c⁺ cells. Using single-cell RNA sequencing (scRNA-seq), the authors

demonstrate that ileal CD11c⁺ cells are microglia-like, and the same subtype of cells is activated in the brain and ileum of PD mice. Moreover, by utilizing mice expressing the photo-convertible protein, Dendra2, they show that CD11c⁺ cells traffic from the brain to the ileum. Together these data provide a novel mechanism of alphaSyn trafficking between the brain and gut.

The data presented suggest a connection in PD mice between the brain and the ileum, mediated by CD11c⁺ macrophages. It would be instructive to show that depletion of CD11c⁺ macrophages eliminate the appearance of pathologic alphaSyn in the ileum. This experiment seems essential to the authors hypothesis.

We thank the reviewer for this suggestion. We were able to acquire 3 appropriately aged B6a.CD11c.DOG animals and their WT littermates. These mice express the diphtheria toxin receptor under the CD11c promoter, allowing for prolonged depletion of CD11c⁺ cells with multiple diphtheria toxin injections (Hochweller et al. 2008). One week after stereotactic injection with an AAV expressing hαSyn, WT and transgenic B6a.CD11c.DOG littermates received IP injections of diphtheria toxin every other day for two weeks. At week three after injection, the intestines of these mice were analyzed for αSyn. The three week timepoint was used, as previous studies have demonstrated that CD11c can only be depleted for two weeks with this mouse strain (Hochweller et al. 2008). Transgenic mice with depleted CD11c demonstrated a drastic reduction in the number of αSyn⁺ villi compared to their WT controls. We believe this data successfully demonstrates that CD11c⁺ cells are responsible for the trafficking of αSyn from the brain to the ileum in the hαSyn PD mouse model. We have added this data to Figure 3I and Extended Figure 2I. The description of these data was added to line 155 as follows:

“To test if CD11c⁺ cells mediate the propagation of αSyn in the hαSyn mouse model, we utilized the transgenic B6a.CD11c.DOG mouse strain. These mice express the diphtheria toxin receptor under the CD11c promoter, allowing for prolonged depletion of CD11c⁺ cells with multiple diphtheria toxin injections (Hochweller et al. 2008). One week after stereotactic injection with an AAV expressing hαSyn, WT and transgenic B6a.CD11c.DOG littermates received IP injections of diphtheria toxin every other day for two weeks. At week three after injection, the intestines of these mice were analyzed for αSyn. The three week timepoint was used, as previous studies have demonstrated that CD11c can only be depleted for two weeks with this mouse strain. Transgenic mice with depleted CD11c demonstrated a drastic reduction in the number of αSyn⁺ villi compared to their WT controls (Fig. 3I & Extended Data Fig. 3K). Taken together these data demonstrate a unique connection in PD mice between the brain and the ileum, mediated by CD11c⁺ macrophages.”

Reviewer #3 (Remarks to the Author):

The study by McFleder and coauthors is an interesting investigation into the dynamics of myeloid cells and their relevance to alpha-synuclein (α-syn) pathology in a mouse model, with important implications for translational immunology as well as human neurodegenerative disease. The authors utilize a broad range of modern techniques which enable the elegant investigation of the research hypothesis. However, some of the key conclusions are not supported by the current data and need either to be demonstrated with additional analysis/data, or to be adjusted.

Basically, the study proposes three key mechanisms: 1) Localized, vector-mediated αSyn overexpression in the brain results in αSyn pathology in the gut independently from vagal transmission 2) CD11c⁺ cells rapidly circulate b/w brain and ileum in physiological and pathological

conditions 3) These CD11c+ cells carry aSyn protein from the brain to the ileum.

The first two mechanisms are somewhat novel, very-well reasoned and backed by the data. However, the third mechanism, which has the highest novelty and potential therapeutic implications, is based on several assumptions that are not fully backed by the data presented.

The first two mechanisms already render the study worth publishing, but the third mechanism needs to be demonstrated better by the data or adjusted to avoid overstatement.

Please find below major and minor points for consideration:

Major point:

1) The main problem is with the concept that the aggregated aSyn in the ileum is produced by the AAV-transduced neurons in the brain, -> released by them -> taken up by the CD11c+ cells -> and transported to the ileum. While this seems highly likely, no data is presented to exclude the possibility that the aggregated aSyn in the ileum is produced by the ileum-residing CD11c+ themselves, because they were also transduced in the brain at the time of injection. If this is the case, this will drastically reduce the relevance to the natural human disease.

There are several relatively easy ways to disprove this:

- Explicitly demonstrate that the mouse CD11c+ cell are not transduced by AAV1/2, either with own data or by citing robust literature studies.
- Demonstrate with RT-qPCR/FISH that the vector and its transcripts are not present in the ileum tissue and best also in the circulating WBCs, for example with a forward primer for human aSyn and a reverse primer for HA-tag/BGH/WPRE. Optimally, this will be shown shortly after treatment, as the authors show that the CD11c+ cells move very quickly between tissues.
- Re-analyze the single-cell data (see below why) to demonstrate that the CD11c+ cells really do not express the transgene themselves

We thank the reviewer for the useful suggestions. We have now reanalyzed our scRNA-Seq data for the full length h α Syn including the HA/WPRE/polyA/BGH portion mentioned below. Here we were able to detect a minimal amount of reads mapping to the h α Syn, however these were from low quality cells (likely contaminating neurons) that did not contain at least 200 features and were therefore filtered out before our downstream analysis. We have included a snapshot of the mapped reads from the brain h α Syn sample before filtering the data (Extended Data Fig. 3H) and a violin plot for the h α Syn gene from all the samples demonstrating no expression of the h α Syn gene (Extended Data Fig. 3G). We believe this extra analysis strengthens our original hypothesis that the CD11c+ cells are not expressing h α Syn but rather trafficking brain-derived h α Syn to the ileum. This is consistent with the previous study of this model, which only demonstrated viral particles transducing neurons and not other cell types (Henrich et al. 2018). We modified the sentence at line 91 to include this citation:

“This “brain-first” PD model preferentially induces localized expression of h α Syn in neurons (Henrich et al. 2018) (Fig. 1A), which is associated with dopaminergic neurodegeneration 5 weeks following the injection (Fig. 1A, 1B & 1C).”

The manuscript has also been edited at line 182 to include the new h α Syn mapping information as follows:

“The h α Syn gene was also examined, where a few mapped reads were detected. However, these reads did not signify expression of the h α Syn in the CD11c+ cells analyzed, as they were present in low quality cells and filtered out after the first quality control step of our analysis pipeline (Extended Data Fig. 3G & 3H). These data provide further evidence that the α Syn in the ileum was not a result of increased ileal α Syn expression but rather transport of protein aggregates from the brain.”

The GFP AAV control provides some support here, however has several problems:

- It would require to show a quantification of GFP-positive cells/villi in Ileum at 1, 5 and 10 weeks after treatment (even if all values are 0), like in Figure 1 E, or even better like in Fig. 2 B with MFI quantification; instead of the single illustrative IHC picture showed in Figure 1 D, third panel. In fact, the low magnification of Fig 1 D does not allow the reader to appreciate if there are indeed no single GFP+ cells. Actually, even if GFP-positive cells are found in the ileum, this will only disprove the specificity of the process to aSyn; it won't necessarily mean that the cells were transduced by the vector. To disprove this, the aforementioned data/analysis/literature is crucial.

As suggested we quantified GFP⁺ cells in the ileum from 1, 5, and 10 week animals. Here only 2 out of 15 animals demonstrated GFP⁺ cells in the ileum, despite sufficient expression of GFP in the SN. Although this demonstrates that GFP can also be trafficked to the ileum, the data still suggest that α Syn is preferentially trafficked at a higher frequency than GFP despite being comparably overexpressed. We have updated the manuscript to reflect this information at line 101 and added the quantification to Fig. 1I.

“Brain-gut trafficking was preferential to α Syn, as expression of green fluorescent protein (GFP) in the SN lead to only minimal amounts of GFP in the ileum (Fig. 1A, 1D & 1I).”

- The authors show that WBC carry aSyn (Ext. Data Fig. 2D & E). It would be interesting to see/exclude if they also carry GFP with the same method. If the WBCs carry GFP, but are not found in the ileum, this will have some interesting implications.

As the reviewer requested, we performed western blot for GFP in WBCs from animals injected with the AAV-GFP virus. Here we were not able to detect GFP in any of the four animals analyzed. As a positive control we have also included a protein sample from the SN of a GFP animal. We have included this information in Extended Data Figure 2F and line 136:

“Indeed, α Syn but not GFP protein could be detected in the circulating white blood cells (WBC) from h α Syn or GFP-injected animals (Extended Data Fig. 2D, E, & F).”

Similarly, the single-cell data can offer some support here. However, its interpretation has one major flaw: The authors state that the lack of aSyn mRNA (Snca gene) in the single-cell data demonstrates that aSyn in CD11c+ cells is taken up and not expressed there (Discussion L256-262). However, they utilized the 10X Genomics 3' GEX kit, which detects almost only the last 400-500 bp of a transcript (3'). Thus, the transgene as described in Kolprich et al. Mol Degener 2010 5:43; will not be assigned to the Snca gene, as only the HA/WPRE/polyA/BGH portion 'behind' human aSyn will be visible to the aligner. I would recommend aligning to a custom mm10 reference including the exact sequence of the transgene transcript with the 5' and 3' sequences and visualizing the alignments on a genomic browser. Also, I think the transgene is very stable for a long time in transduced cells, but what if the cells do not express it at 5 weeks after treatment (time point of the single-cell data) just because it was already degraded/inactivated? Could the authors comment on this?

The reviewer is correct that transgene production by AAVs is very stable and can induce long term expression of genes. In our model we have seen h α Syn production extending out as far as 16 weeks (Karikari et al. 2022). We therefore believe that the five week time point is not too late to visualize

h α Syn expression in CD11c⁺ cells. As described above, we have reanalyzed the sequencing data as requested and indeed do not find expression of the h α Syn in the CD11c⁺ cells from any of the analyzed organs. The manuscript has also been edited at line 182 to include the new h α Syn mapping information as follows:

“The h α Syn gene was also examined, where a few mapped reads were detected. However, these reads did not signify expression of the h α Syn in the CD11c⁺ cells analyzed, as they were present in low quality cells and filtered out after the first quality control step of our analysis pipeline (Extended Data Fig. 3G & 3H). These data provide further evidence that the α Syn in the ileum was not a result of increased ileal α Syn expression but rather transport of protein aggregates from the brain.”

Finally, even if the authors cannot unequivocally solve the major concern, the hypothesis remains novel, highly likely and worthy; but then at least the conclusions need to be adjusted. For example, L247 “Through the use of an HA-tag, our study definitively shows that α Syn molecules traffic out of the brain and to the gut. “ is an overstatement if the authors do not show data or literature that excludes the possibility that the CD11c⁺ cells were expressing the transgene themselves, possibly even only after leaving the brain.

We thank the reviewer for pointing out the novelty of our study. Due to the valuable suggestion of this reviewer, we were able to include data demonstrating that the CD11c⁺ cells do not express h α Syn. Hence we are content with leaving our conclusion unmodified.

Minor points:

2) Fig 2B: Looking at the distribution of the data in 2B, aSyn can be considered aggregating ALSO already at 1 and 5 weeks, what are the p-values here? Related to this, line 263-264 from the discussion “Despite being a brain-first model of PD, the h α Syn animals demonstrated α Syn aggregations in the 264 ileum prior to significant accumulations in the brain.” is an exaggeration, since there is obviously an increase in MFI of PK-resistant aSyn MFI in SNpc at 1 week post treatment, even if the p-value is not below 0.05. To ‘prove’ the lack of difference, the authors would have to report the post-hoc statistical power β for detecting a difference of at least the magnitude which is seen at 10 weeks.

This is an important observation. The p-values for the PK-resistant α Syn at week 1 and week 5 are 0.5281 and 0.1998 respectively. Although the data are not significant, the PK-resistant α Syn in the brain is trending upwards and analysis of more brains may be necessary to reach significance. We have therefore revised the sentence at line 302 to read as follows:

“Despite being a brain-first model of PD, the h α Syn animals demonstrated α Syn aggregations in the ileum at early stages of disease.”

We have also edited the results section at line 118 with this information:

“These data demonstrate that in our brain-first model of PD, pathological forms of α Syn are apparent at early timepoints in the intestines.”

3) Results L118-L220 “This suggests that expression of the mutated human α Syn leads to aggregation of endogenous α Syn, likely explaining why mostly endogenous α Syn propagates to the ileum in this

model.”: What is the justification that it shows exactly that? It could be e.g. that the inflammation caused by the model results in the expression of the endogenous aSyn.

The reviewer is correct that it could have also been the inflammation that caused the endogenous α Syn to aggregate. With this sentence we were trying to portray that the brain aggregates in this model consist mainly of endogenous HA- α Syn, which explains why the ileum also contains mostly HA- α Syn. We have revised the sentence at line 120 to better reflect this:

“This suggests that protein aggregations in the brains of h α Syn animals consist mainly of endogenous α Syn, likely explaining why mostly endogenous α Syn propagates to the ileum in this model.”

4) Related to the major concern, I suggest using more concise wording: for the presence of aSyn protein from the brain in the ileum, use “presence”, not expression, which can be understood here as gene expression.

We thank the reviewer for the suggestion. We have updated the manuscript to only mention expression when referring to the h α Syn in the SN at line 76 (removed expression).

5) L136: “...tended to co-localize...” is misleading, as it is obvious in the IHC that there are many other aSyn-staining cells that are not CD11c (green) and vice versa. Maybe re-phrase?

In this sentence we were referring to the α Syn⁺ cells in the ileum. To be more clear, we have rephrased the sentence at line 139 as follows:

“Intriguingly, α Syn tended to colocalize with CD11c⁺ cells in the ileum (Fig. 3A, top panel).”

7) The legend of Fig. 3 D specifies red and green in the IHC staining, are these the correct colors? It is hard to appreciate in the figure.

The definition of red (CD11c) and green (alpha-synuclein) IHC staining is correct. Nevertheless, since this may be hard to see on the small panels and especially the typically turquoise looking histogreen may be difficult to distinguish from the hematoxylin-stained blue nuclei, we have updated the figure 3D and 3H to include high resolution pictures and thereby make the colors more apparent.

8) Seems like there are two very distinct groups of PD patients in Fig. 3 E, can the authors comment on this?

We thank the reviewer for this interesting comment. The figure is indeed suggestive of two groups of PD patients with either low or high α Syn⁺CD11c⁺ cell numbers. An elevated cell number principally reflects the resorption activity following degeneration of Lewy body containing neurons, which varies from patient to patient and depends on the spatiotemporal distribution of neuronal loss within an individual SN. We investigated whether the patients of the two clusters differed concerning the number of residual neurons, CD4- and CD8-associated immunological activity, general resorption activity measured by CD68 immunohistochemistry as well as clinical parameters such as generalized systemic inflammation (e.g. sepsis), other diseases and cause of death. None of the stated parameters showed significant differences between the groups. Since the number of subjects in each cluster is limited, we feel that it is not warranted to interpret the data as two distinct groups, but rather rely on future results of larger cohorts.

9) Fig. 4 D – The most profound change of cell type proportions is in the spleen, which seems contra-intuitive to the main hypothesis, why could that be?

Based on our previous publication, we know that the h α Syn PD mouse model does not only demonstrate inflammatory changes in the brain but also in peripheral immune organs such as the spleen and the cervical lymph node (Karikari et al. 2022). This is also consistent with the peripheral immune changes described in patients, where changes in circulating immune cells have been observed by several papers (Yuhua Chen et al. 2015). We believe these peripheral immune reactions outside of the intestines does not contradict the main hypothesis that a specific subset of CD11c⁺ cells traffic α Syn from the brain to the ileum. This may in turn lead to activation of other cell types that then cause immune dysregulation in other organs. We have updated the manuscript at line 207 to acknowledge these changes in the spleen:

“Consistent with our previous data of systemic immune dysregulation in PD, splenic CD11c⁺ cells also underwent shifts in their cluster distributions. Expansion of TRM 1 cells, however, was specific to the brain and the ileum.”

10) A part of the last sentence of Fig. 4 legend is missing.

We thank the reviewer for pointing this out. We have edited the last sentence as follows:

“Sequencing data is from the pooled organs of 5 EV and 5 h α Syn animals harvested 5 weeks following viral vector injection.”

11) Fig. 6D: The authors state that there are no Dendra-Red-positive cells in the LNs and Spleen; here the quantification (percent of Dendra-Red-positive cells) in all 8 animals is more appropriate than one representative plot - obviously these are the 8 animals that are shown in Fig 6. C, of which 4 animals have very low number of photoconverted cells also in the ileum and brain. Do the 3 animals with high frequency (> 0.1 %) of photoconverted cells in the brain also show 0 % converted cells in LNs and spleen?

Indeed, it was very interesting to see that even when our efficiency of photoconversion in the brain was high we did not observe a significant increase in photoconverted cells in the cervical lymph nodes or spleen of the same animals. The representative LN and spleen images in Fig. 6 is from an animal with high efficiency of photoconversion in the brain (EV + stim in Fig. 6A). We have added this information to the figure legend and have also included a quantification of all 8 animals in the Extended Data Fig. 5C.

12) GO analysis – Since these cells were all gated on their high(er) expression of CD11c, which an important part of leucocyte migration, and often co-regulated with other leucocyte migration molecules, it comes to no surprise that comparing gene sets from this cells to the whole mouse genome returns enrichment of leucocyte migration terms. It would be good if the authors show as extended data if the top 10 GO terms of the other populations markers do not include leucocyte migration terms to prove their point that this is a characteristic of the TRM1/TRM2/M1 cells.

This is a very astute observation made by the reviewer. We have therefore performed GO analysis on all 11 CD11c⁺ populations and confirmed that only the Macrophage 1 cluster was enriched with migration-related terms (6/10) in comparison to all other clusters which only contained a maximum

of 2. We have included the top 10 GO terms in the supplementary table and added this information into the manuscript at line 235 as follows:

“Indeed, six out of the top ten GO terms in the Macrophage 1 cluster were migration-related, which was in sharp contrast to the other CD11c⁺ clusters which contained a maximum of two migration-related terms (Supplemental Table 1).”

We hope that we could address all reviewer concerns in the revised manuscript and look forward to a hopefully positive response.

Best regards

Chi Wang Ip

References

Challis, Collin, Acacia Hori, Timothy R. Sampson, Bryan B. Yoo, Rosemary C. Challis, Adam M. Hamilton, Sarkis K. Mazmanian, Laura A. Volpicelli-Daley, and Viviana Gradinaru. 2020. ‘Gut-Seeded α -Synuclein Fibrils Promote Gut Dysfunction and Brain Pathology Specifically in Aged Mice’. *Nature Neuroscience* 23 (3): 327–36. <https://doi.org/10.1038/s41593-020-0589-7>.

Chen, Yan, Yu Guo, Payam Gharibani, Jie Chen, Florin M. Selaru, and Jiande D. Z. Chen. 2021. ‘Transitional Changes in Gastrointestinal Transit and Rectal Sensitivity from Active to Recovery of Inflammation in a Rodent Model of Colitis’. *Scientific Reports* 11 (1): 8284. <https://doi.org/10.1038/s41598-021-87814-7>.

Chen, Yuhua, Benquan Qi, Wenfang Xu, Bo Ma, Li Li, Qiming Chen, Weidong Qian, Xiaolin Liu, and Hongdang Qu. 2015. ‘Clinical Correlation of Peripheral CD4⁺-Cell Sub-Sets, Their Imbalance and Parkinson’s Disease’. *Molecular Medicine Reports* 12 (4): 6105–11. <https://doi.org/10.3892/mmr.2015.4136>.

Clarner, Tim, Katharina Janssen, Lara Nellessen, Martin Stangel, Thomas Skripuletz, Barbara Krauspe, Franz-Martin Hess, et al. 2015. ‘CXCL10 Triggers Early Microglial Activation in the Cuprizone Model’. *Journal of Immunology (Baltimore, Md.: 1950)* 194 (7): 3400–3413. <https://doi.org/10.4049/jimmunol.1401459>.

Ghaisas, Shivani, Monica R. Langley, Bharathi N. Palanisamy, Somak Dutta, Kirthi Narayanaswamy, Paul J. Plummer, Souvarish Sarkar, et al. 2019. ‘MitoPark Transgenic Mouse Model Recapitulates the Gastrointestinal Dysfunction and Gut-Microbiome Changes of Parkinson’s Disease’. *Neurotoxicology* 75 (December): 186–99. <https://doi.org/10.1016/j.neuro.2019.09.004>.

Groh, Janos, Konrad Knöpper, Panagiota Arampatzi, Xidi Yuan, Lena Lößlein, Antoine-Emmanuel Saliba, Wolfgang Kastenmüller, and Rudolf Martini. 2021. ‘Accumulation of Cytotoxic T Cells in the Aged CNS Leads to Axon Degeneration and Contributes to Cognitive and Motor Decline’. *Nature Aging* 1 (4): 357–67. <https://doi.org/10.1038/s43587-021-00049-z>.

Henrich, Martin Timo, Fanni Fruzsina Geibl, Bolam Lee, Wei-Hua Chiu, James Benjamin Koprach, Jonathan Michael Brotchie, Lars Timmermann, Niels Decher, Lina Anita Matschke, and Wolfgang Hermann Oertel. 2018. ‘A53T- α -Synuclein Overexpression in Murine Locus Coeruleus Induces Parkinson’s Disease-like Pathology in Neurons and Glia’. *Acta Neuropathologica Communications* 6 (1): 39. <https://doi.org/10.1186/s40478-018-0541-1>.

Hochweller, Kristin, Jörg Striegler, Günter J. Hämmerling, and Natalio Garbi. 2008. 'A Novel CD11c.DTR Transgenic Mouse for Depletion of Dendritic Cells Reveals Their Requirement for Homeostatic Proliferation of Natural Killer Cells'. *European Journal of Immunology* 38 (10): 2776–83. <https://doi.org/10.1002/eji.200838659>.

REVIEWERS' COMMENTS

Reviewer #1 (Remarks to the Author):

In the manuscript by McFleder and colleagues, the authors present the novel observation that exogenous alpha synuclein introduced into the brain can be trafficked to the gut by CD11c+ immune cells. The significance of this trafficking on disease pathology and the identification of the immune populations that transport alpha synuclein are key questions. The authors identify intestinal dysfunction (increased fecal output and shorter transit time), that correspond with brain and intestinal (ileum) inflammation. The authors examine the CD11c+ populations through flow cytometry, immunohistochemistry (IHC), and single-cell RNA sequencing (scRNA-seq). These data provide good characterization of the types of CD11c+ cells in the brain and ileum, indicating F4/80+CD11c+ macrophages are accumulating in the disease state. Finally, the authors demonstrate that CD11c+ cells can migrate from the brain to the gut using transgenic mice with photoconvertible fluorescent protein expression. It is surprising not to see trafficking from the brain to the brain-draining lymph nodes or spleen, as these are important immune cell hubs for interactions of myeloid cell populations with T cells. However, in the field, there are many unanswered questions about immune cell trafficking to and from the brain. Overall, the data presented provide some new areas to explore in the trafficking of immune cells and alpha synuclein transport, and it will be interesting to see, in future studies, the molecular mechanisms involved. The data presented were obtained with appropriate technique and statistical analysis. In the introduction and discussion, the manuscript references previous literature appropriately. The authors have addressed all the previous concerns and suggestions.

Reviewer #2 (Remarks to the Author):

Although the experiment the authors performed is somewhat limited to show that depletion of CD11c+ macrophages eliminate the appearance of pathologic alphaSyn in the ileum, it is supportive of their hypothesis. The authors should acknowledge the limitations of their supporting evidence and discuss future definitive experiments that are required to support to the notion CD11c+ cells traffic from the brain to the ileum such as KI mouse studies, etc.

Reviewer #3 (Remarks to the Author):

In the revised manuscript, McFleder and colleagues have added valuable additional data and adjusted the data interpretation to support the conclusions and fairly depict how the data supports the hypothesis. The authors have adequately addressed my concerns and in my view that of the other reviewers. My only remaining concern is that the reader is not sufficiently made aware of the strength/weakness of evidence that haSyn is really not expressed in the CD11c cells. The newly added scRNA plots in ext.fig 3G,H are somewhat informative, but without a cell type (e.g. in the violin plot) as a positive control to show that haSyn can be robustly detected if it was really expressed, it remains not very convincing if really no cell type expressed the transgene haSyn or it is just not easily detectable in the 3'GEX assay. I would recommend including a sentence concerning this in the discussion.

Response to reviewer's Comments

Reviewer #1 (Remarks to the Author):

In the manuscript by McFleder and colleagues, the authors present the novel observation that exogenous alpha synuclein introduced into the brain can be trafficked to the gut by CD11c+ immune cells. The significance of this trafficking on disease pathology and the identification of the immune populations that transport alpha synuclein are key questions. The authors identify intestinal dysfunction (increased fecal output and shorter transit time), that correspond with brain and intestinal (ileum) inflammation. The authors examine the CD11c+ populations through flow cytometry, immunohistochemistry (IHC), and single-cell RNA sequencing (scRNA-seq). These data provide good characterization of the types of CD11c+ cells in the brain and ileum, indicating F4/80+CD11c+ macrophages are accumulating in the disease state. Finally, the authors demonstrate that CD11c+ cells can migrate from the brain to the gut using transgenic mice with photoconvertible fluorescent protein expression. It is surprising not to see trafficking from the brain to the brain-draining lymph nodes or spleen, as these are important immune cell hubs for interactions of myeloid cell populations with T cells. However, in the field, there are many unanswered questions about immune cell trafficking to and from the brain. Overall, the data presented provide some new areas to explore in the trafficking of immune cells and alpha synuclein transport, and it will be interesting to see, in future studies, the molecular mechanisms involved. The data presented were obtained with appropriate technique and statistical analysis. In the introduction and discussion, the manuscript references previous literature appropriately. The authors have addressed all the previous concerns and suggestions.

We thank the reviewer for their positive comments and previous helpful suggestions which helped to strengthen the manuscript.

Reviewer #2 (Remarks to the Author):

Although the experiment the authors performed is somewhat limited to show that depletion of CD11c+ macrophages eliminate the appearance of pathologic alphaSyn in the ileum, it is supportive of their hypothesis. The authors should acknowledge the limitations of their supporting evidence and discuss future definitive experiments that are required to support the notion CD11c+ cells traffic from the brain to the ileum such as KI mouse studies, etc.

We appreciate the comment from the reviewer and have added the following sentence to line 292 in the manuscript:

Depletion of CD11c⁺ cells seemed to decrease transport of α Syn to the ileum of h α Syn mice, however this finding is limited by the low n number and transient decrease in CD11c. Future studies which invoke long-term depletion of CD11c⁺ cells through the use of CD11c-Cre animals are required to definitively demonstrate that CD11c⁺ cells are required for α Syn transport.

Reviewer #3 (Remarks to the Author):

In the revised manuscript, McFleder and colleagues have added valuable additional data and adjusted the data interpretation to support the conclusions and fairly depict how the data supports

the hypothesis. The authors have adequately addressed my concerns and in my view that of the other reviewers. My only remaining concern is that the reader is not sufficiently made aware of the strength/weakness of evidence that haSyn is really not expressed in the CD11c cells. The newly added scRNA plots in ext.fig 3G,H are somewhat informative, but without a cell type (e.g. in the violin plot) as a positive control to show that haSyn can be robustly detected if it was really expressed, it remains not very convincing if really no cell type expressed the transgene haSyn or it is just not easily detectable in the 3'GEX assay. I would recommend including a sentence concerning this in the discussion.

We acknowledge the limitation of the sequencing analysis for h α Syn in our CD11c⁺ scRNA-seq results. Because we did not sequence neurons in this experiment there is no positive control to demonstrate the ability to detect h α Syn mRNA with our method. We have added the following passage to the discussion at line 289 to better inform the reader:

Although the scRNA-seq data is limited by the lack of a positive cell type expressing h α Syn, the absence of this mRNA in our CD11c⁺ cell scRNA-seq data supports a transfer of α Syn protein from neurons to CD11c⁺ cells.

We hope these changes adequately address the reviewers concerns and look forward to a hopefully positive response.